# PF-LRM: Pose-Free Large Reconstruction Model for Joint Pose and Shape Prediction

**Peng Wang**[*]
Adobe Research & HKU
totoro97@outlook.com

**Hao Tan**
Adobe Research
hatan@adobe.com

**Sai Bi**
Adobe Research
sbi@adobe.com

**Yinghao Xu**[*]
Adobe Research & Stanford
yhxu@stanford.edu

**Fujun Luan**
Adobe Research
fluan@adobe.com

**Kalyan Sunkavalli**
Adobe Research
sunkaval@adobe.com

**Wenping Wang**
Texas A&M University
wenping@tamu.edu

**Zexiang Xu**
Adobe Research
zexu@adobe.com

**Kai Zhang**
Adobe Research
kaiz@adobe.com

## Abstract

We propose a **P**ose-**F**ree **L**arge **R**econstruction **M**odel (*PF-LRM*) for reconstructing a 3D object from a few *unposed* images even with little visual overlap, while simultaneously estimating the relative camera poses in ∼1.3 seconds on a single A100 GPU. *PF-LRM* is a highly scalable method utilizing self-attention blocks to exchange information between 3D object tokens and 2D image tokens; we predict a coarse point cloud for each view, and then use a differentiable Perspective-n-Point (PnP) solver to obtain camera poses. When trained on a huge amount of multi-view posed data of ∼1M objects, *PF-LRM* shows strong cross-dataset generalization ability, and outperforms baseline methods by a large margin in terms of pose prediction accuracy and 3D reconstruction quality on various unseen evaluation datasets. We also demonstrate our model's applicability in downstream text/image-to-3D task with fast feed-forward inference. Our project website is at: https://totoro97.github.io/pf-lrm.

## 1 Introduction

3D reconstruction is a classical computer vision problem, with applications spanning imaging, perception, and computer graphics. While both traditional photogrammetry (Barnes et al., 2009; Schönberger et al., 2016; Furukawa & Ponce, 2009) and modern neural reconstruction methods (Mildenhall et al., 2020; Wang et al., 2021; Yariv et al., 2021) have made significant progress in high-fidelity geometry and appearance reconstruction, they rely on having images with calibrated camera poses as input. These poses are typically computed using a Structure-from-Motion (SfM) solver (Schonberger & Frahm, 2016; Snavely et al., 2006).

SfM assumes dense viewpoints of the scene where input images have sufficient overlap and matching image features. However, this is not applicable in many cases, e.g., e-commerce applications, consumer capture scenarios and dynamic scene reconstruction problems, where adding more views incurs a higher cost and thus the captured views tend to be *sparse* and have a *wide baseline* (i.e., share little overlap). In such circumstances, SfM solvers become unreliable and tend to fail. As a result, (neural) reconstruction methods, including sparse methods (Niemeyer et al., 2022; Deng et al., 2022; Long et al., 2022; Zhou & Tulsiani, 2023; Wang et al., 2023) that require accurate camera poses, cannot be reliably used for such applications.

In this work, we present *PF-LRM*, a category-agnostic method for jointly predicting camera poses and reconstructing object shape and appearance (represented using a triplane NeRF (Chan et al.,

---

[*]This work was done while the authors were interns at Adobe Research.

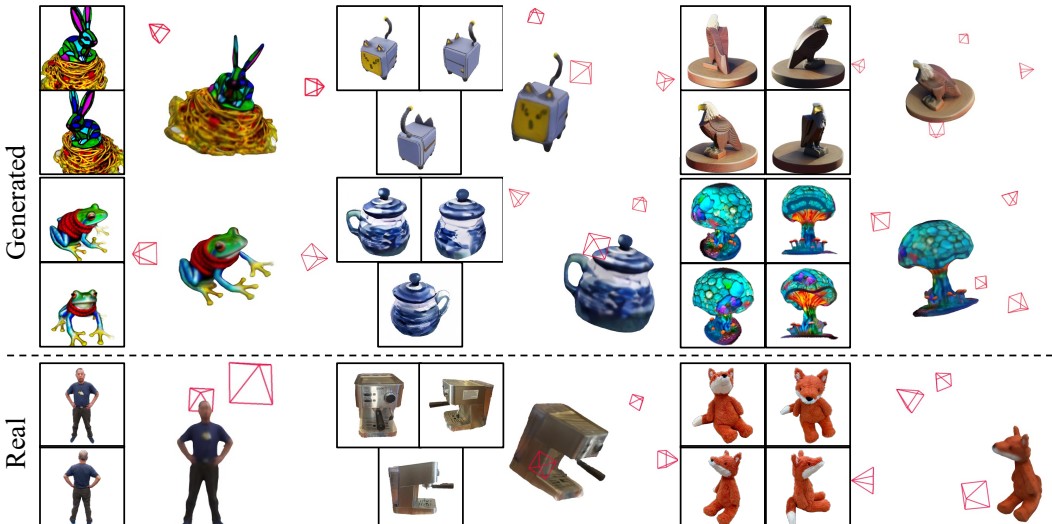

Figure 1: (Top block) To demonstrate our model's generalizability to unseen in-the-wild images, we take 2-4 unposed images (without background) from prior/concurrent 3D-oriented generation work, and use our *PF-LRM* to jointly reconstruct the NeRF and estimate relative poses in a feed-forward manner. Sources of generated images: Row 1 (left-to-right), Magic3D (Lin et al., 2023b), Zero-1-to-3 (Liu et al., 2023a), MVDream (Shi et al., 2023b); Row 2 (left-to-right) DreamFusion (Poole et al., 2022), SyncDreamer (Liu et al., 2023b), Zero123++ (Shi et al., 2023a). (Bottom block) We also show our model's generalizability on real captures. Source of real images: Row 1 (left-to-right), HuMMan Dataset (Cai et al., 2022), RelPose++ (Lin et al., 2023a), our phone capture. For additional test cases, please refer to our project page.

2022; Peng et al., 2020)). As shown in Fig. 1, our approach can robustly reconstruct accurate poses and realistic 3D objects using as few as 2–4 sparse input images from diverse input sources. The core of our approach is a novel scalable single-stream transformer model that computes self-attention over the union of two token sets: a set of 2D multi-view image tokens and a set of 3D triplane NeRF tokens, allowing for comprehensive information exchange across all 2D and 3D tokens. We use the final NeRF tokens, contextualized by 2D images, to represent a triplane NeRF and supervise it by a rendering loss. On the other hand, we use the final image patch tokens contextualized by NeRF tokens to predict coarse point clouds that are subsequently used to solve per-view camera poses.

Unlike previous methods that regress pose parameters from images directly, we estimate the 3D object points corresponding to 2D image patch centers from their individual patch tokens. These points are supervised by the NeRF geometry in an online manner during training and enable accurate pose estimation using a differentiable Perspective-n-Point (PnP) solver (Chen et al., 2022b). In essence, we transform the task from per-view pose prediction into per-patch 3D surface point prediction, which is more suitable for our single-stream transformer that is designed for token-wise operations, leading to more accurate results than direct pose regression.

*PF-LRM* is a large transformer model with ∼590 million parameters trained on large-scale multi-view posed renderings from Objaverse (Deitke et al., 2023) and real-world captures from MVImgNet (Yu et al., 2023) that cover ∼1 million objects in total, without direct 3D supervision. Despite being trained under the setting of 4 input views, it generalizes well to unseen datasets and can handle a variable number of 2–4 unposed input images during test time (see Fig. 1), achieving state-of-the-art results for both pose estimation and novel view synthesis in the case of very sparse inputs, outperforming baseline methods (Jiang et al., 2022; Lin et al., 2023a) by a large margin.

## 2 RELATED WORK

**NeRF from sparse posed images.** The original NeRF technique (Mildenhall et al., 2020) required hundreds of posed images for accurate reconstruction. Recent research on sparse-view NeRF reconstruction has proposed either regularization strategies (Wang et al., 2023; Niemeyer et al., 2022;

Yang et al., 2023; Kim et al., 2022) or learning priors from extensive datasets (Yu et al., 2021; Chen et al., 2021; Long et al., 2022; Ren et al., 2023; Zhou & Tulsiani, 2023; Irshad et al., 2023; Li et al., 2023; Xu et al., 2023). These approaches still assume precise camera poses for every input image; however determining camera poses given such sparse-view images is non-trivial and off-the-shelf camera estimation pipelines (Schonberger & Frahm, 2016; Snavely et al., 2006) tend to fail. In contrast, our method efficiently reconstructs a NeRF (Chan et al., 2022; Chen et al., 2022a; Peng et al., 2020) from sparse views without any camera pose inputs; moreover, our method is capable of recovering the unknown relative camera poses during inference time with the learned shape prior.

**Structure from Motion.** Structure-from-Motion (SfM) techniques (Schonberger & Frahm, 2016; Snavely et al., 2006; Mohr et al., 1995) find 2D feature matches across views, and then solve for camera poses and sparse 3D structure from these 2D correspondences. These methods work well in the presence of sufficient visual overlap between nearby views and adequate discriminative features. However, when the input views are extremely sparse, for instance, when there are only 4 images looking from the front-, left-, right-, back- side of an object, it becomes very challenging to match features across views due to the lack of sufficient overlap, even with modern learning-based feature extractors (DeTone et al., 2018; Dusmanu et al., 2019; Revaud et al., 2019) and matchers (Sarlin et al., 2020; 2019; Liu et al., 2021). In contrast, our method relies on the powerful learned shape prior from a large amount of data to successfully register the cameras in these challenging scenarios.

**Neural pose prediction from RGB images.** A series of methods (Lin et al., 2023a; Rockwell et al., 2022; Cai et al., 2021) have sought to address this issue by directly regressing camera poses through network predictions. Notably, these methods do not incorporate 3D shape information during the camera pose prediction process. We demonstrate that jointly reasoning about camera pose and 3D shape leads to significant improvement over these previous methods that only regress the camera pose. SparsePose (Sinha et al., 2023), FORGE (Jiang et al., 2022) and FvOR (Yang et al., 2022) implement a two-stage prediction pipeline, initially inferring coarse camera poses and coarse shapes and then refining these pose predictions (through further network evaluations (Sinha et al., 2023) or per-object optimizations (Jiang et al., 2022; Yang et al., 2022)) jointly with 3D structures. Our method employs a single-stage inference pipeline to recover both camera poses and 3D NeRF reconstructions simultaneously. To predict camera poses, we opt not to regress them directly as in prior work. Instead, we predict a coarse point cloud in the scene coordinate frame (Shotton et al., 2013) for each view from their image patch tokens; these points, with image patch centers, establish 3D-2D correspondences, and allow us to solve for poses using a differentiable PnP solver (Chen et al., 2022b; Brachmann et al., 2017). This is in contrast to solving poses from frame-to-frame scene flows (3D-3D correspondences) used by FlowCam (Smith et al., 2023), and better suits the sparse view inputs with little overlap. Moreover, our backbone model is a simple transformer-based model that is highly scalable; hence it can be trained on massive multi-view posed data of diverse and general objects to gain superior robustness and generalization. This distinguishes us from the virtual correspondence work (Ma et al., 2022) that's designed specifically for human images.

## 3 METHOD

Given $N$ images $\{I_i | i = 1, .., N\}$ with unknown camera poses capturing a 3D object, our goal is to reconstruct the object's 3D model and estimate the pose of each image. In particular, we designate one input image (i.e., $I_1$) as a *reference* view, and predict a triplane NeRF and camera poses of other images relative to the reference view. This is expressed by

$$T, y_2, ..., y_N = \text{PF-LRM}(I_1, ..., I_N), \tag{1}$$

where $T$ is the triplane NeRF defined in the coordinate frame of the reference view 1 and $y_2,...,y_N$ are the predicted camera poses of view $2, ..., N$ relative to view 1.

We achieve this using a transformer model as illustrated in Fig. 2. Specifically, we tokenize both input images and NeRF tokens, and apply a single-stream multimodal transformer (Chen et al., 2020; Li et al., 2019) to process the concatenation of NeRF tokens and image patch tokens with self-attention layers (Sec. 3.1). The output NeRF tokens represent a triplane NeRF for differentiable rendering, and the output image patch tokens are used to estimate per-view coarse point cloud for pose estimation with a differentiable PnP solver (Sec. 3.3).

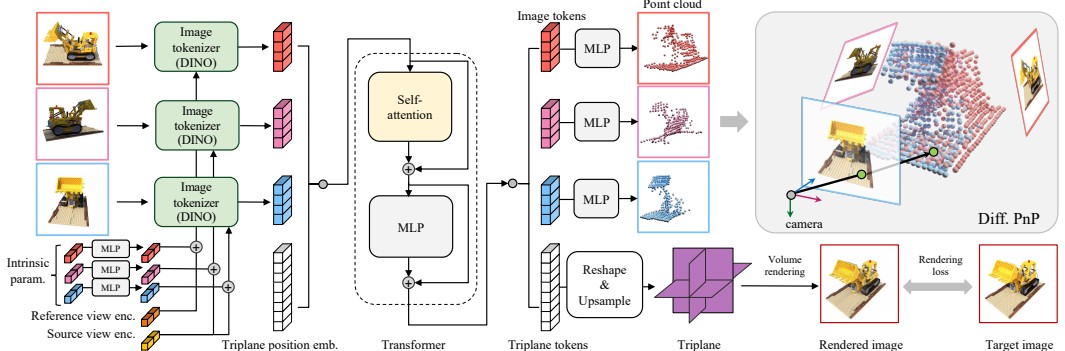

Figure 2: **Overview of our pipeline.** We use a transformer model to reconstruct a triplane NeRF while simultaneously estimating the relative camera poses of all source input views with respect to the reference input image. The triplane tokens are supervised with a rendering loss at novel viewpoints using ground-truth camera poses. For pose estimation, instead of directly regressing the camera poses, we map the patch tokens to a coarse 3D geometry (top right), where we predict a 3D point from each patch token corresponding to the patch center. We then use a differentiable PnP solver to obtain the camera poses from these predicted 3D-2D correspondences (Sec. 3.3).

## 3.1 SINGLE-STREAM TRANSFORMER

**Image tokenization, view encoding, intrinsics conditioning**. We use the pretrained DINO (Caron et al., 2021) Vision Transformer (Dosovitskiy et al., 2020) to tokenize our input images. Each input image of resolution $H \times W$ is tokenized into $M = H/16 \times W/16$ tokens.

To distinguish reference image tokens from source image tokens, we use two additional learnable features, $\boldsymbol{v}_r$ and $\boldsymbol{v}_s$, as view encoding vectors – one $\boldsymbol{v}_r$ for the reference view ($i = 1$) and another $\boldsymbol{v}_s$ for all other source views ($i = 2, .., N$). These view encoding vectors are used to modulate the per-view image patch tokens within DINO, allow our model to estimate camera poses (i.e., extrinsic parameters) relative to the reference view. In addition, to make our model aware of input cameras' intrinsics, we use a shared MLP to map each view's intrinsics $[f_x, f_y, c_x, c_y]$ to a conditioning vector $\boldsymbol{i} \in \mathbf{R}^{768}$; hence we have $\boldsymbol{i}_r, \boldsymbol{i}_i, i = 2, ..., N$ for reference and source views, respectively. We then pass the addition of each view's view encoding and intrinsics conditioning vectors to the newly-added adaptive layer norm block inside each transformer block (self-attention + MLP), following prior work (Hong et al., 2023; Peebles & Xie, 2022; Huang & Belongie, 2017).

**Triplane tokenization and position embedding.** We tokenize a triplane $\boldsymbol{T}$ into $3 \times H_T \times W_T$ tokens, where $H_T, W_T$ denote triplane height and width. We additionally learn a triplane position embedding $\boldsymbol{T}_{pos}$ consisting of $3 \times H_T \times W_T$ position markers for triplane tokens; they are mapped to the target triplane tokens by our transformer sourcing information from input image tokens.

**Single-stream transformer**. The full process of this single-stream transformer can be written as

$$\boldsymbol{T}, \{\boldsymbol{a}_{i,j} | i = 1, .., N; j = 1, ..., M\} = \text{PF-LRM}(\boldsymbol{T}_{pos}, \boldsymbol{I}_1, ..., \boldsymbol{I}_N, \boldsymbol{v}_r, \boldsymbol{v}_s). \quad (2)$$

Here $\boldsymbol{a}_{i,j}$ represents the token of the $j^{\text{th}}$ patch at view $i$, and PF-LRM is a sequence of transformer layers. Each transformer layer is composed of a self-attention layer and a multi-layer perceptron (MLP), where both use residual connections. We concatenate the image tokens and the triplane tokens as Transformer's input as shown in Fig. 2. The output triplane tokens $\boldsymbol{T}$ and image tokens $\boldsymbol{a}_{i,j}$ are used for NeRF rendering and per-view pose prediction, which we will discuss later. While inspired by LRM (Hong et al., 2023) and its follow-ups (Li et al., 2023; Xu et al., 2023), our model design is different and has its own unique philosophy. We adopt a single-stream architecture where information exchange is mutual between image tokens and NeRF tokens because we predict both a coherent NeRF and per-view coarse geometry used for camera estimation (detailed later in Sec. 3.3). In contrast, prior work adopts an encoder-decoder design where NeRF tokens source unidirectional information from image tokens using cross-attention layers.

## 3.2 NeRF Supervision via Differentiable Volume rendering

The shape and appearance are supervised via differentiable volume rendering, as done in (Mildenhall et al., 2020; Chan et al., 2022). This process is expressed by

$$C = \sum_{k=1}^{K} \tau_{k-1}(1 - \exp(-\sigma_k \delta_k))c_k, \quad \tau_k = \exp(-\sum_{k'=1}^{k} \sigma_{k'}\delta_{k'}), \quad \sigma_k, c_k = \mathrm{MLP}_{\boldsymbol{T}}(\boldsymbol{T}(\boldsymbol{x}_k)). \quad (3)$$

Here, $C$ is the rendered pixel color, $\sigma_k$ and $c_k$ are volume density and color decoded from the triplane NeRF $\boldsymbol{T}$ at the location $\boldsymbol{x}_k$ on the marching ray through the pixel, and $\tau_k$ ($\tau_0$ is defined to be 1) and $\delta_k$ are the volume transmittance and step size; $\boldsymbol{T}(\boldsymbol{x}_k)$ represents the features that are bilinearly sampled and concatenated from the triplane at $\boldsymbol{x}_k$, and we apply an MLP network $\mathrm{MLP}_{\boldsymbol{T}}$ to decode the density and color used in volume rendering.

We supervise our NeRF rendering with L2 and VGG-based LPIPS (Zhang et al., 2018) loss:

$$L_C = \gamma'_C \, \|C - C_{gt}\|^2 + \gamma''_C \, L_{lpips}(C, C_{gt}), \quad (4)$$

where $C_{gt}$ is the ground-truth color, and $\gamma'_C, \gamma''_C$ are loss weights. In practice, we render crops of size $h \times w$ for each view to compute the rendering loss $L_C$.

## 3.3 Pose prediction via Differentiable PnP Solver

We estimate relative camera poses from the per-view image patch tokens contextualized by the NeRF tokens. Note that a straightforward solution is to directly regress camera pose parameters from the image tokens using an MLP decoder; however, such a naïve solution lacks 3D inductive biases and, in our experiments (See Tab. 7), often leads to limited pose accuracy. Therefore, we propose to predict per-view coarse geometry (in the form of a sparse point cloud, i.e., predicting one 3D point for each patch token) that is supervised to be consistent with the NeRF geometry, allowing us to obtain the camera poses with a PnP solver given the 3D-2D correspondences from the per-patch predicted points and patch centers.

In particular, from each image patch token output by the transformer $\boldsymbol{a}_{i,j}$, we use an MLP to predict a 3D point and the prediction confidence:

$$\boldsymbol{p}_{i,j}, \alpha_{i,j}, w_{i,j} = \mathrm{MLP}_{\boldsymbol{a}}(\boldsymbol{a}_{i,j}), \quad (5)$$

where $\boldsymbol{p}_{i,j}$ represents the 3D point location on the object seen through the central pixel of the image patch, $\alpha_{i,j}$ is the pixel opacity that indicates if the pixel covers the foreground object, and $w_{i,j}$ is an additional confidence weight used to determine the point's contribution to the PnP solver.

Note that in training, where the ground-truth camera poses are known, the central pixel's point location and opacity can also be computed from a NeRF as done in previous work (Mildenhall et al., 2020). This allows us to enforce the consistency between the per-patch point estimates and the triplane NeRF geometry with following losses:

$$L_{\boldsymbol{p}} = \sum_{i,j} \|\boldsymbol{p}_{i,j} - \bar{\boldsymbol{x}}_{i,j}\|^2, \quad L_\alpha = \sum_{i,j} (\alpha_{i,j} - (1 - \bar{\tau}_{i,j}))^2, \quad (6)$$

where $\bar{\boldsymbol{x}}$ and $\bar{\tau}$ are computed along the pixel ray (marched from the ground-truth camera poses) using the volume rendering weights in Eqn. 3 by

$$\bar{\boldsymbol{x}} = \sum_{k=1}^{K} \tau_{k-1}(1 - \exp(-\sigma_k \delta_k))\boldsymbol{x}_k, \quad \bar{\tau} = \tau_K = \exp(-\sum_{k'=1}^{K} \sigma_{k'}\delta_{k'}), \quad (7)$$

where $\tau_k, \sigma_k, \delta_k$ follow the definition of Eqn. 3. Essentially, as we only use multi-view posed images to train our model without accessing 3D ground-truth, we distill the geometry of our learnt NeRF reconstruction to supervise our per-view point cloud prediction in an online manner. This online distillation is critical to stabilize the differentiable PnP loss mentioned later in Eq. 11; without it we find that the training tends to diverge in our experiments.

Once $\boldsymbol{p}_{i,j}$ and $\alpha_{i,j}$ are estimated, we can compute each pose $y_i$ with a weighted PnP solver as:

$$\underset{y_i=[R_i,t_i]}{\arg\min} \frac{1}{2} \sum_{j=1}^{M} \xi(y_i, \boldsymbol{p}_{i,j}, \beta_{i,j}, \boldsymbol{q}_{i,j}), \tag{8}$$

$$\xi(y_i, \boldsymbol{p}_{i,j}, \beta_{i,j}, \boldsymbol{q}_{i,j}) = \beta_{i,j} \|\mathcal{P}(R_i \cdot \boldsymbol{p}_{i,j} + t_i) - \boldsymbol{q}_{i,j}\|^2, \tag{9}$$

$$\beta_{i,j} = \alpha_{i,j} w_{i,j}, \tag{10}$$

where $\boldsymbol{q}_{i,j}$ is the 2D central pixel location of the patch, $[R_i, t_i]$ are the rotation and translation of the pose $y_i$, $\mathcal{P}$ is the projection function with camera intrinsics involved. Here, the predicted opacity values are used to prevent the non-informative background points from affecting the pose prediction.

However, computing the solution of PnP is a non-convex problem prone to local minimas. Therefore, we further apply a robust differentiable PnP loss, proposed by EPro-PnP (Chen et al., 2022b) [1], to regularize our pose prediction, leading to much more accurate results (See Tab. 7). This loss is expressed by

$$L_{y_i} = \frac{1}{2} \sum_j \xi(y_i^{\text{gt}}, \boldsymbol{p}_{i,j}, \beta_{i,j}) + \log \int \exp\left(-\frac{1}{2} \sum_j \xi(y_i, \boldsymbol{p}_{i,j}, \beta_{i,j})\right) \mathrm{d}y_i, \tag{11}$$

where the first term minimizes the reprojection errors of the predicted points with the ground-truth poses and the second term minimizes the reprojection errors with the predicted pose distribution using Monte Carlo integral; we refer readers to the EPro-PnP paper (Chen et al., 2022b) for details.

### 3.4 Loss functions and implementation details

**Loss.** Combining all losses (Eqn. 4,6,11), our final training objective is

$$L = L_C + \gamma_{\boldsymbol{p}} L_{\boldsymbol{p}} + \gamma_\alpha L_\alpha + \gamma_y \sum_{i=2}^{M} L_{y_i}, \tag{12}$$

where $L_C$ represents the rendering loss and $\gamma_{\boldsymbol{p}}$, $\gamma_\alpha$, $\gamma_y$ are the weights for individual loss terms related to per-view coarse geometry prediction, opacity prediction and differentiable PnP loss.

**Implementation details.** Our single-stream transformer model consists of 36 self-attention layers. We predict triplane of shape $H_T = W_T = 64, D_T = 32$. In order to decrease the tokens used in transformer, the triplane tokens used in transformer is $3072 = 3\times32\times32$ and is upsampled to 64 with de-convolution, similar to LRM (Hong et al., 2023). We set the loss weights $\gamma_C', \gamma_C'', \gamma_{\boldsymbol{p}}, \gamma_\alpha, \gamma_y$ to $1, 2, 1, 1, 1$, respectively. For more details, please refer to Sec. A.3 of the appendix.

## 4 Experiments

### 4.1 Experimental settings

**Training & evaluation datasets.** Our model only requires multi-view posed images to train. We use a mixture of multi-view posed renderings from Objaverse (Deitke et al., 2023) and posed real captures from MVImgNet (Yu et al., 2023) for training. To evaluate our model's cross-dataset generalization capability, we utilize a couple of datasets, including OmniObject3D (Wu et al., 2023), Google Scanned Objects (GSO) (Downs et al., 2022), Amazon Berkeley Objects (ABO) (Collins et al., 2022), Common Objects 3D (CO3D) (Reizenstein et al., 2021), and DTU (Aanæs et al., 2016). For OmniObject3D, GSO, ABO datasets, we randomly choose 500 objects for assessing our model's performance. For CO3D dataset, we use the 400 held-out captures provided by Rel-Pose++ (Lin et al., 2023a), which covers 10 object categories. For DTU dataset, we take the 15 objects with manually annotated masks provided by IDR (Yariv et al., 2020); for each object, we randomly select 8 different combinations of four input views, resulting in a total of 120 different testing inputs. Additional details can be found in the appendix.

**Baselines.** As our *PF-LRM* can do joint pose and shape estimation, we evaluate its performance against baselines on both tasks. For the pose estimation task, we compare *PF-LRM* with

---

[1]We take the public implementation in https://github.com/tjiiv-cprg/EPro-PnP.

Table 1: Quantitative results of pose prediction task on novel datasets. On OmniObject3D, GSO, ABO where background is white in the rendered data, we evaluate the *w/o bg* variant of RelPose++, while on others where real captures contain background, we evaluate its *w/ bg* variant.

| Dataset | Method | R. error ↓ | Acc.@15° ↑ | Acc.@30° ↑ | T. error ↓ |
|---|---|---|---|---|---|
| *OmniObject3D* | FORGE | 71.06 | 0.071 | 0.232 | 0.726 |
| | HLoc (F. rate 99.6%) | 98.65 | 0.083 | 0.083 | 1.343 |
| | RelPose++ (w/o bg) | 69.22 | 0.070 | 0.273 | 0.712 |
| | Ours | **6.32** | **0.962** | **0.990** | **0.067** |
| *GSO* | FORGE | 103.81 | 0.012 | 0.056 | 1.100 |
| | HLoc (F. rate 97.2%) | 97.12 | 0.036 | 0.131 | 1.199 |
| | RelPose++ (w/o bg) | 107.49 | 0.037 | 0.098 | 1.143 |
| | Ours | **3.99** | **0.956** | **0.976** | **0.041** |
| *ABO* | FORGE | 105.23 | 0.014 | 0.059 | 1.107 |
| | HLoc (F. rate 98.8%) | 94.84 | 0.067 | 0.178 | 1.302 |
| | RelPose++ (w/o bg) | 102.30 | 0.060 | 0.144 | 1.103 |
| | Ours | **16.27** | **0.865** | **0.885** | **0.150** |
| *CO3D* | FORGE | 77.74 | 0.139 | 0.278 | 1.181 |
| | HLoc (F. rate 89.0%) | 55.87 | 0.288 | 0.447 | 1.109 |
| | RelPose++ (w/ bg) | 28.24 | 0.748 | 0.840 | 0.448 |
| | Ours | **15.53** | **0.850** | **0.899** | **0.242** |
| *DTU* | FORGE | 78.88 | 0.046 | 0.188 | 1.397 |
| | HLoc (F. rate 47.5%) | 11.84 | 0.725 | 0.915 | 0.520 |
| | RelPose++ (w/ bg) | 41.84 | 0.369 | 0.657 | 0.754 |
| | Ours | **10.42** | **0.900** | **0.951** | **0.187** |

FORGE (Jiang et al., 2022), RelPose++ (Lin et al., 2023a), and the SfM-based method HLoc (Sarlin et al., 2019; Schonberger & Frahm, 2016). We also compare with FORGE in terms of the reconstruction quality. SRT (Sajjadi et al., 2022) is geometry-free and does not directly predict shapes like us; hence we did not compare with it due to this clear distinction in problem scopes.

**Metrics.** We use the pair-wise related pose errors as our metric as in RelPose++ (Lin et al., 2023a). We also report the percentage of image pairs with relative rotation errors below 15° and 30°. The translation part of the predicted relative pose is measured by its absolute difference from the ground-truth. We evaluate the reconstruction quality by comparing renderings of reconstructed NeRF using both *predicted* input-view poses and *ground-truth* novel-view poses against the ground-truth. We report the PSNR, SSIM and LPIPS (Zhang et al., 2018) metrics for measuring the image quality. We use 4 images as inputs for each object when comparing the performance of different methods.

## 4.2 EXPERIMENT RESULTS

### 4.2.1 POSE PREDICTION QUALITY

As shown in Tab. 1, our model achieves state-of-the-art results in pose estimation accuracy given highly sparse input images on unseen datasets including OmniObject3d, GSO and ABO, consistently outperforming baselines by a large margin. We observe that HLoc has a very high failure rate (more than 97%) on these very sparse inputs, because matching features is very hard in this case; this also highlights the difficulty of pose prediction under extremely sparse views, and the contributions of this work to the area. On the held-out CO3D test set provided by RelPose++, our rotation error is much smaller than FORGE and HLoc, and 1.8x smaller than RelPose++ (*w/ bg*). Note that on the CO3D, our method along with FORGE and HLoc are all tested on input images with background removed. The inaccurate foreground masks provided by CO3D can negatively influence these methods' performance; In addition, RelPose++ is trained on the CO3D training set while the other methods, including ours, are not. On DTU dataset where none of the methods are trained on, we achieve over 4× rotation error reduction than RelPose++ (*w/ bg*) and FORGE, showing much better generalization capability.

We attribute our model's success to the joint prediction of both camera poses and object shapes, where the synergy of the two tasks is exploited by the self-attention mechanism. Prior methods like RelPose++ fail to utilize this synergy, as they solve for poses directly from images without reconstructing the 3D object. FORGE introduced shape prior into the pose estimation process,

Table 2: Quantitative evaluations comparing with FORGE (Jiang et al., 2022) on novel view synthesis quality.

| Method | OmniObject3D | | | Google Scanned Objects | | | Amazon Berkeley Objects | | |
|---|---|---|---|---|---|---|---|---|---|
| | PSNR ↑ | SSIM ↑ | LPIPS ↓ | PSNR ↑ | SSIM ↑ | LPIPS ↓ | PSNR ↑ | SSIM ↑ | LPIPS ↓ |
| | Evaluate renderings of our predicted NeRF at novel-view poses | | | | | | | | |
| FORGE | 17.95 | 0.800 | 0.215 | 11.43 | 0.754 | 0.760 | 10.92 | 0.669 | 0.325 |
| Ours | **23.02** | **0.877** | **0.083** | **25.04** | **0.879** | **0.096** | **26.23** | **0.887** | **0.097** |
| | Evaluate renderings of our predicted NeRF at our predicted poses | | | | | | | | |
| FORGE | 19.03 | 0.829 | 0.189 | 11.90 | 0.760 | 0.202 | 11.32 | 11.32 | 0.209 |
| Ours | **27.27** | **0.916** | **0.054** | **27.01** | **0.914** | **0.0645** | **27.19** | **0.894** | **0.083** |

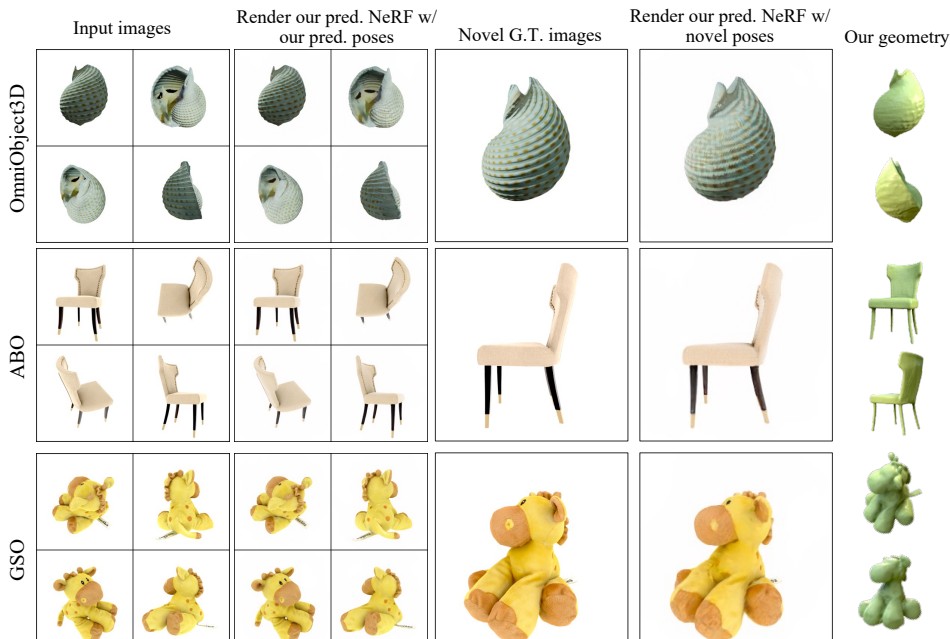

Figure 3: Cross-dataset generalization of *PF-LRM* to OmniObject3D, GSO and ABO datasets.

but its training process is composed of six stages, which is fragile and hard to scale up. We also acknowledge that if one can successfully scale up the training data of the baseline RelPose++ and FORGE, their performance can also be improved compared to training with limited data. We show one such experiment where we re-train RelPose++ on Objaverse in appendix A.7; however, our model, trained on the same dataset, still outperforms this re-trained baseline by a wide margin.

### 4.2.2 RECONSTRUCTION QUALITY

We evaluate the view synthesis quality by comparing the quality of our reconstructed NeRF to that of FORGE (Jiang et al., 2022). To isolate the influence of inaccurate masks on measuring the view synthesis quality, we evaluate on OmniObject3D, GSO, and ABO datasets and compare with the FORGE baseline. We use the sample input settings as in the pose prediction task. As shown in Tab. 2, *PF-LRM* achieves an average PSNR of 24.8 on the three datasets, while FORGE's average PSNR is only 13.4. This shows that our model generalizes well in terms of reconstruction quality on unseen datasets while FORGE does not. On the other hand, we think there's an important objective to fulfill in the task of joint pose and NeRF prediction - the predicted NeRF, when rendered at predicted poses, should match well the input unposed images. This objective is complementary to novel view quality and requires accurate predictions of both poses and NeRF. We show in Tab. 2 that our *PF-LRM* performs much better than FORGE with high PSNR values.

In general, our model learns a generic shape prior effectively from massive multi-view datasets, thanks to its scalable single-stream transformer design. FORGE's multi-stage training is challenging

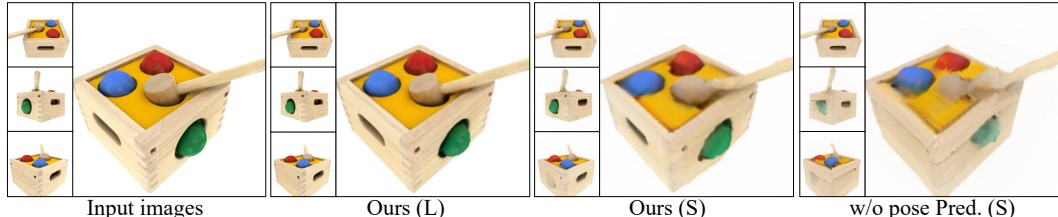

| Input images | Ours (L) | Ours (S) | w/o pose Pred. (S) |

Figure 4: Ablation studies. Note that for a fair comparison of different ablation variants we render out our reconstructed NeRF using the same *ground-truth* poses corresponding to input images.

to scale due to error accumulation across stages. Fig. 3 qualitatively shows the high-quality NeRF reconstruction and accurate pose prediction from our model. We also demonstrate extracted meshes from our reconstructed NeRF; the meshes are extracted by first rendering out 100 RGBD images uniformly distributed in a sphere and then fusing them using RGBD fusion (Curless & Levoy, 2023).

## 4.3 ABLATION STUDIES

Table 3: Ablation study of model size and training objectives on the GSO dataset.

| Setting | R. error | Acc.@$15°$ | Acc.@$30°$ | T. error | $PSNR_{g.t.}$ | $PSNR_{pred.}$ |
|---|---|---|---|---|---|---|
| Ours (L) | **2.46** | **0.976** | **0.985** | **0.026** | **29.42** | **28.38** |
| Ours (S) | 13.08 | 0.848 | 0.916 | 0.135 | 23.80 | 22.82 |
| w/o NeRF Pred. (S) | 111.89 | 0.000 | 0.000 | 1.630 | - | - |
| w/o pose Pred. (S) | - | - | - | - | 22.48 | - |

In the ablation studies, we train our models with different settings on the synthetic Objaverse dataset (Deitke et al., 2023) and evaluate on GSO dataset (Downs et al., 2022) to isolate the influence of noisy background removals. For better energy efficiency, we conduct ablations mostly on a smaller version of our model, dubbed as Ours (S). It has 24 self-attention layers with 1024 token dimension, and is trained on 8 A100 GPUs for 20 epochs (∼100k iterations), which takes around 5 days. In addition, to show the scaling law with respect to model sizes, we train a large model (Ours (L)) on 128 GPUs for 100 epochs (∼70k iterations).

**Using smaller model.** 'Ours (L)' outperforms the smaller one 'Ours (S)' by a great margin in terms of pose prediction accuracy and NeRF reconstruction quality, as shown in Tab. 3 and Fig. 4.

**Removing NeRF prediction.** Here we evaluate two different settings: 1) using differentiable PnP for pose prediction as described in Sec. 3.3; 2) using MLP to directly predict poses from the concatenated patch features. For 1), we notice that the training becomes very unstable and diverges in this case, as we find that our point loss (Eqn. 6; relying on NeRF prediction) helps stabilize the PnP loss (Eqn. 11). For 2), we find that the predicted pose is almost random, as shown in Tab. 3;

**Removing pose prediction.** We find that jointly predicting pose helps the model learn better 3D reconstruction with sharper textures, as shown in Tab. 3 and Fig. 4 (comparing 'w/o pose Pred. (S)' and 'Ours (S)'). This could be that by forcing the model to figure out the correct spatial relationship of input views, we reduce the uncertainty and difficulty of shape reconstruction.

## 5 CONCLUSION

In this work, we propose a large reconstruction model based on the transformer architecture to jointly estimate camera parameters and reconstruct 3D shapes in the form of NeRF. Our model employs self-attention to allow triplane tokens and image patch tokens to communicate information with each other, leading to improved NeRF reconstruction quality and robust per-patch surface point prediction for solving poses using a differentiable PnP solver. Trained on multi-view posed renderings of the large-scale Objaverse and real MVImgNet datasets, our model outperforms baseline methods by a large margin in terms of pose prediction accuracy and reconstruction quality.

**Ethics Statement.** This paper proposes a reconstruction model that can convert multi-view images to the 3D shapes. This technique can be used to reconstruct images with humans. However, the current shape resolution is still relatively low and cannot get accurate reconstructions of the face and hand regions. The model is trained to be a deterministic model and thus, it is hard to reproduce the data used in training. Users can use this model to reconstruct 3D shapes from images where the shape might have commercial IP. This model also utilizes compute for training that is significantly larger than previous 3D reconstruction models. This could potentially lead to a trend of pursuing large reconstruction models in the 3D domain, leading to environmental concerns similar to the current trend of large language models.

**Reproducibility Statement.** We have elucidated our model design in the paper including the training architecture (transformer in Sec. 3.1, NeRF rendering in Sec. 3.2) the losses (pose loss in Sec. 3.3 and final loss in Sec. 3.4). The training details are shown in Sec. 3.4 and further extended in Appendix. We have also pointed to the exact implementation of the Diff. PnP method in Sec. 3.3 to resolve uncertainty over the detailed implementation. Lastly, we will be involve in discussions regarding the implementation details of our paper.

**Acknowledgement.** We want to thank Nathan Carr, Duygu Ceylan, Paul Guerrero, Chun-Hao Huang, and Niloy Mitra for discussions on this project. We thank Yuan Liu for helpful discussions on pose estimation. Peng Wang was partially supported by the Hong Kong ITC under the InnoHK initiative and Ref. T45-205/21-N of Hong Kong RGC.

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

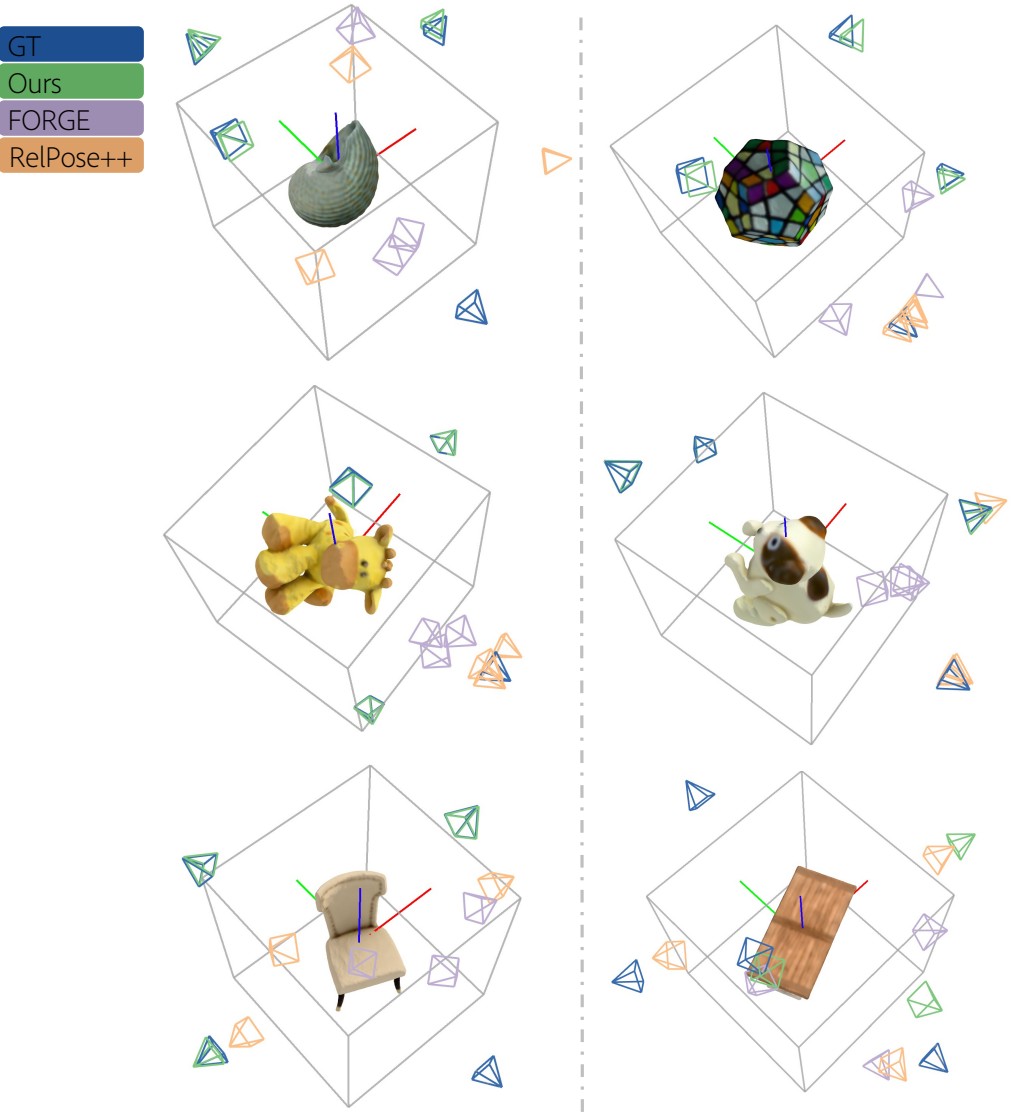

Figure 5: Predicted poses from our method align much more closely with the ground-truth than those from baseline methods including FORGE (Jiang et al., 2022), RelPose++ (Lin et al., 2023a). The bottom right camera is the GT reference camera pose, and we did not show its corresponding prediction cameras.

# A APPENDIX

## A.1 VISUAL COMPARISONS OF PREDICTED CAMERA POSES

In Fig. 5, we present visual comparisons of the predicted camera poses with our method and baseline methods. We can see that it's common for baseline methods FORGE (Jiang et al., 2022) and RelPose++ (Lin et al., 2023a) to make predictions significantly deviating from the ground truth and in some situations, their predicted camera poses can be even on the opposite side. In contrast, our predicted poses closely align with the ground truth consistently.

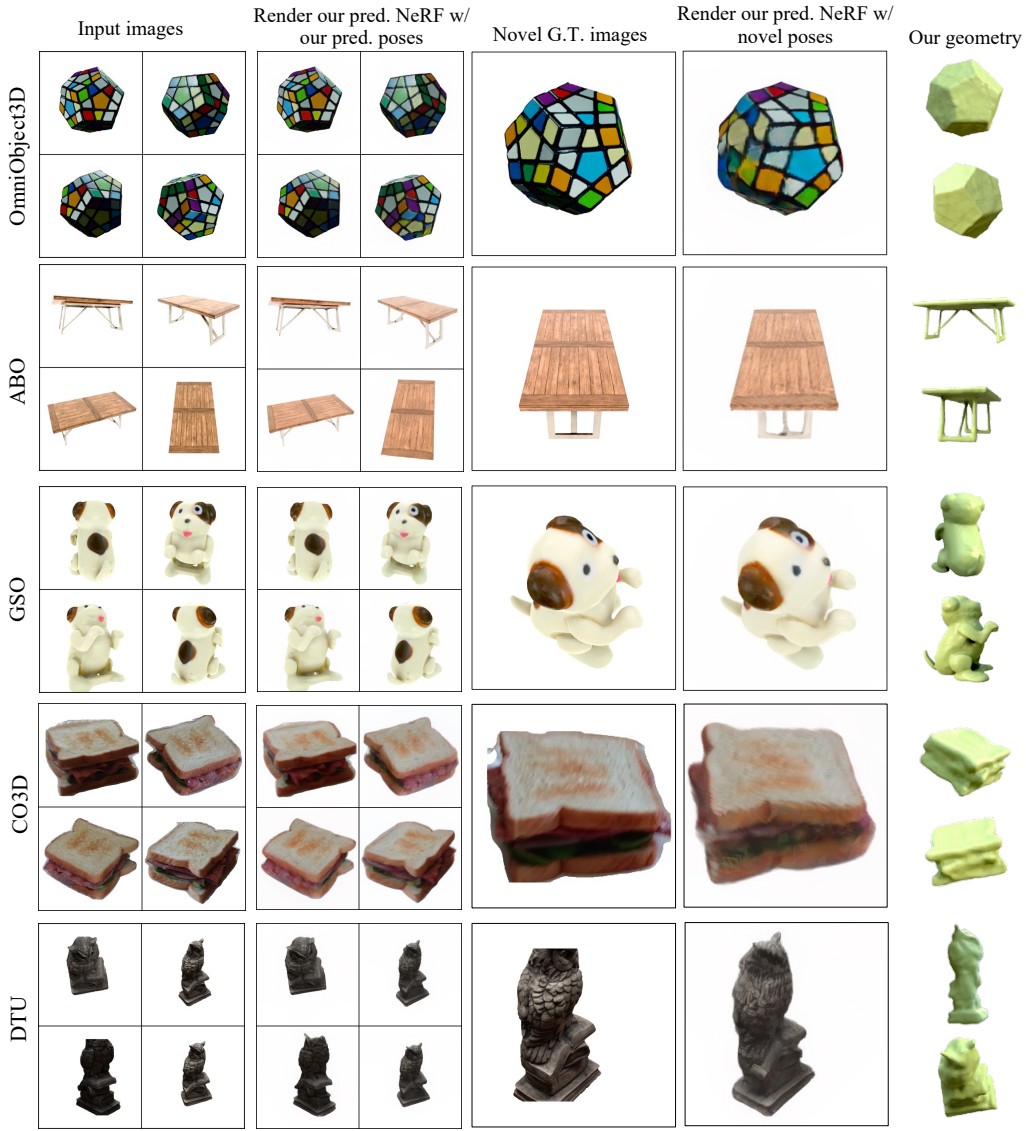

Figure 6: Additional qualitative results of our model's cross-dataset generalization to unseen OmniObject3D (Wu et al., 2023), GSO (Downs et al., 2022), ABO (Collins et al., 2022), CO3D (Reizenstein et al., 2021), and DTU (Aanæs et al., 2016) datasets.

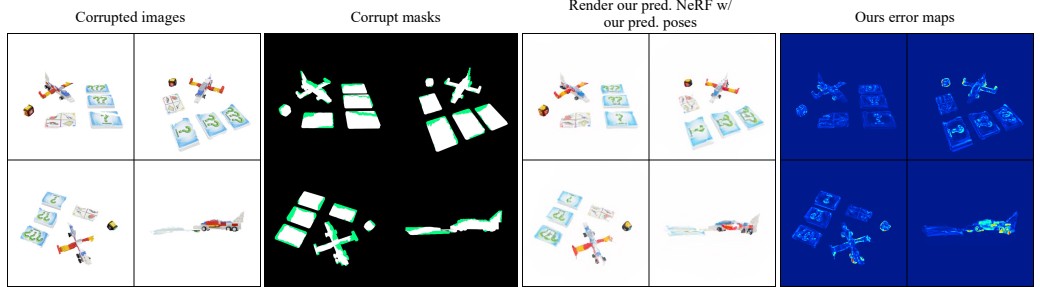

|  | Corrupted images | Corrupt masks | Render our pred. NeRF w/ our pred. poses | Ours error maps |

Figure 7: Our *PF-LRM* is robust to small mask segmentation errors.

**Variable number of input views.** Our model naturally supports variable number of input views as a result of the transformer-based architecture. We test our model's performance on variable number of input images on the 500 selected GSO objects (see Sec. 4.1). As shown in Tab. 4, with decreased number of views, we observe a consistent drop in reconstruction quality and pose prediction quality, but the performance degradation is acceptable. For pose evaluation, we only evaluate the relative pose errors of the first two views for fair comparison. $PSNR_{input}$ reflects how well our model's predicted NeRF and poses can explain the input images, while the $PSNR_{all}$ is an aggregated metrics including both input views and held-out novel views (we have 4 views in total for each object).

Table 4: Inference on variable number of input views on unseen GSO dataset using our *PF-LRM* trained on 4 views (no re-training or fine-tuning is involved). For pose evaluation, we only evaluate the relative pose errors of the first two views for fair comparison.

| #Views | R. error | Acc.@15° | Acc.@30° | $PSNR_{input}$ | $PSNR_{all}$ |
|---|---|---|---|---|---|
| 4 | **4.19** | **0.956** | **0.974** | 27.76 | **27.76** |
| 3 | 5.83 | 0.946 | 0.962 | 27.59 | 26.76 |
| 2 | 10.38 | 0.886 | 0.924 | 27.35 | 24.87 |
| 1 | - | - | - | **29.27** | 21.56 |

**Robustness to imperfect segmentation masks.** In this experiment we add noises on the input segmentation masks by adding different levels of elastic transform (Simard et al., 2003). As shown in Tab. 5, we can see that our model is robust to certain level of noise, but its performance drops significantly when the masks are very noisy. This is also aligned with the observation that the inaccurate masks provided by CO3D (Reizenstein et al., 2021) can harm our model's performance on it. Note that $PSNR_{g.t.}$ reflects how well renderings of our predicted NeRF using ground-truth input poses match the input images, while $PSNR_{pred}$ measures how well renderings of our predicted NeRF using our pose predictions match the inputs.

Table 5: Inference on images with varying level of segmentation mask errors on unseen GSO dataset using our *PF-LRM*.

| Noise level | R. error | Acc.@15° | Acc.@30° | T. error | $PSNR_{g.t.}$ | $PSNR_{pred.}$ |
|---|---|---|---|---|---|---|
| 0 | **2.46** | **0.976** | **0.985** | **0.026** | **29.42** | **28.38** |
| 1 | 4.84 | 0.951 | 0.968 | 0.050 | 27.19 | 26.84 |
| 2 | 7.15 | 0.921 | 0.946 | 0.075 | 26.25 | 26.26 |
| 3 | 10.34 | 0.881 | 0.916 | 0.106 | 25.515 | 25.567 |
| 4 | 14.13 | 0.844 | 0.894 | 0.141 | 24.934 | 24.975 |

**Robustness to novel environment lights.** We evaluate our model's robustness to different environment lights in Table 6. The evaluations are conducted in 100 object samples from GSO dataset (Downs et al., 2022). Our model shows consistent results under different lighting conditions. We also qualitatively shows our model robustness to different illuminations in Fig. 8. Note that $PSNR_{g.t.}$ reflects how well renderings of our predicted NeRF using ground-truth input poses

match the input images, while $\text{PSNR}_{pred}$ measures how well renderings of our predicted NeRF using our poses predictions match the inputs.

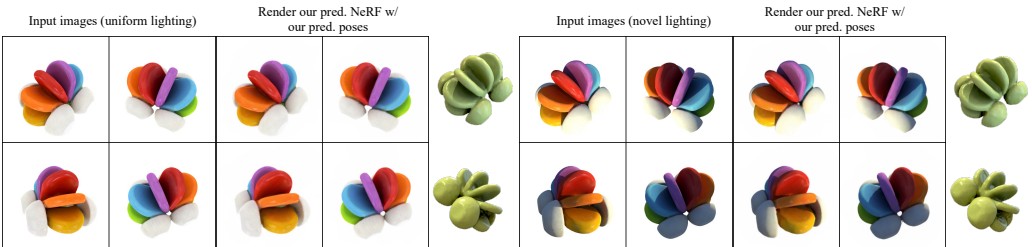

Figure 8: Our *PF-LRM* is robust to illumination changes.

Table 6: Evaluation results on GSO data with different novel environment lights. The evaluations are conducted in 100 objects samples. Note our synthesized multi-view training images are rendered using uniform light. Our method can generalize well to novel environment lights.

| Method | R. error | Acc.@15° | Acc.@30° | T. error | $\text{PSNR}_{\text{g.t.}}$ | $\text{PSNR}_{\text{pred.}}$ |
|--------|----------|----------|----------|----------|--------|---------|
| Sunset | 2.40 | 0.968 | 0.983 | 0.027 | 27.56 | 26.74 |
| Sunrise | 2.22 | 0.985 | 0.993 | 0.024 | 27.17 | 26.21 |
| Studio | 2.82 | 0.983 | 0.992 | 0.029 | 27.31 | 26.69 |
| Uniform | 3.94 | 0.968 | 0.972 | 0.040 | 27.50 | 26.80 |

**Ablations of pose prediction methods.** We illustrate the effectiveness of our differentiable PnP pose prediction method in Tab. 7 by replacing it with alternative solutions. The first line 'diff. PnP' is our model with small config, i.e., 'Ours (S)'. For other lines, we replace the 'diff. PnP' with other alternatives. 'MLP pose (CLS token)' takes the [CLS] token of each view in the last transformer layer to a MLP to predict pose, and supervise pose with a quaternion loss and a translation loss. Although this model can predict plausible reconstructions and poses, its performance is far worse than our full model where we use a differentiable PnP solver to predict poses. We argue that this is because pose prediction has multiple local minimas, and the regression-based pose loss is more prone to such local minimas, compared with the EPro-PnP solver (Chen et al., 2022b) we use in this work. 'MLP pose (Patch tokens)' take concatenated patch-wise features to a MLP for predicting pose. It aims to leverage more dense patch token information in the pose prediction. The performance of this variant is roughly the same as 'MLP pose (CLS token)'. 'non-diff. PnP' removes the differentiable PnP prediction and only use the losses $L_p$ and $L_\alpha$. This way, we have a set of 3D-2D correspondences weighted by predicted opacity that are passed to a PnP solver for getting the poses. We find that this variant leads to worse performance than its differentiable PnP counterpart, due to the lack of learning proper confidence of 3D-2D correspondences. Note that $\text{PSNR}_{g.t.}$ reflects how well renderings of our predicted NeRF using ground-truth input poses match the input images, while $\text{PSNR}_{pred}$ measures how well renderings of our predicted NeRF using our pose predictions match the inputs. We observe that worse pose predictions tend to lead to worse reconstruction quality, as shown by the positive correlation between pose accuracy and $\text{PSNR}_{g.t.}$ scores in Tab. 7.

### A.3 ADDITIONAL IMPLEMENTATION DETAILS

Our model uses a pre-trained DINO ViT as our image encoder. We specifically take DINO-ViT-B/16 with a patch size of $16 \times 16$ and a 12-layer transformer of width $D = 768$. We bilinearly interpolate the original positional embedding to the desired image size. For each view, its view encoding vector and camera intrinsics are first mapped to a modulation feature, and then passed to the adaptive layer norm block (Hong et al., 2023; Peebles & Xie, 2022; Huang & Belongie, 2017) to predict scale and bias for modulating the intermediate feature activations inside each transformer block (self-attention + MLP) of the DINO ViT (Caron et al., 2021). Take the reference view as an example; its modulation feature $\mathbf{m}_r$ is defined as:

$$\mathbf{m}_r = \text{MLP}^{\text{intrin.}}([f_x, f_y, c_x, c_y]) + \boldsymbol{v}_r, \tag{13}$$

where $f_x, f_y, c_x, c_y$ are camera intrinsics, and $\boldsymbol{v}_r$ is the view encoding vector. We then use the modulation feature $\mathbf{m}_r$ in the same way as the camera feature in LRM (Li et al., 2023).

Table 7: Ablation study of different pose prediction methods on the GSO data (Downs et al., 2022). Ablations are conducted with our small model, i.e., 'Ours (S)'. Compared with our method of predicting per-view coarse geometry followed by differentiable PnP (Chen et al., 2022b), the MLP-based pose prediction method conditioning on either the per-view CLS token or the concatenated patch tokens perform much worse due to the lack of explicit geometric inductive bias (either 3D-2D correspondences or 2D-2D correspondences) in pose registrations. Besides, we also find that differentiable PnP learns to weigh the 3D-2D correspondences induced from the per-view predicted coarse geometry properly, resulting a boost in pose estimation accuracy.

| Setting | R. error | Acc.@$15°$ | Acc.@$30°$ | T. error | PSNR$_{g.t.}$ | PSNR$_{pred.}$ |
|---------|----------|-----------|-----------|----------|------|------|
| diff. PnP (our default setting) | **13.08** | **0.848** | **0.916** | **0.135** | **23.80** | **22.82** |
| MLP pose (CLS token) | 25.32 | 0.655 | 0.809 | 0.264 | 22.27 | 19.80 |
| MLP pose (Patch tokens) | 21.60 | 0.688 | 0.836 | 0.230 | 22.02 | 19.76 |
| non-diff. PnP | 22.03 | 0.570 | 0.814 | 0.236 | 23.56 | 18.65 |

| | | Ours (S) | Ours (L) |
|---|---|---|---|
| DINO Encoder | Image resolution | 256×256 | 512×512 |
| | Patch size | 16 | 16 |
| | Att. Layers | 12 | 12 |
| | Attention channels | 768 | 768 |
| | View encoding | 768 | 768 |
| | Intrinsics-cond. MLP layers | 5 | 5 |
| | Intrinsics-cond. MLP width | 768 | 768 |
| | Intrinsics-cond. MLP act. | GeLU | GeLU |
| Transformer | Triplane tokens | $32 \times 32 \times 3$ | $32 \times 32 \times 3$ |
| | Attention channels | 1024 | 1024 |
| | Attention heads | 16 | 16 |
| | Attention layers | 24 | 36 |
| | Triplane upsample | 1 | 2 |
| | Triplane shape | $32 \times 32 \times 3 \times 32$ | $64 \times 64 \times 3 \times 32$ |
| Renderer | Rendering patch size | 64 | 128 |
| | Ray-marching steps | 64 | 128 |
| | MLP layers | 5 | 5 |
| | MLP width | 64 | 64 |
| | Activation | ReLU | ReLU |
| Point MLP | MLP layers | 4 | 4 |
| | MLP width | 512 | 512 |
| | Activation | GeLU | GeLU |
| Traininig | Learning rate | 4e-4 | 4e-4 |
| | Optimizer | AdamW | AdamW |
| | Betas | (0.9, 0.95) | (0.9, 0.95) |
| | Warm-up steps | 3000 | 3000 |
| | Batch size per GPU | 16 | 8 |
| | #GPUS | 8 | 128 |

Table 8: Configuration of our models.

We then concatenate the image tokens with the learnable triplane position embedding to get a long token sequence, which is used as input to the single-stream transformer. We use the multi-head attention with head dimension 64. During rendering, the three planes are queried independently and the three features are concatenated as input of the NeRF MLP to get the RGB color and NeRF density. For per-view geometry prediction used for PnP solver, we use the image tokens output by the transformer with MLP layers to get the point predictions, the confidence predictions, and also the alpha predictions.

In our experiments we have models with two different sizes. In the ablation studies as described in Sec. 4.3, the 'Ours (S)' model has 24 self-attention layers, while the 'Ours (L)' model has 36 self-attention leyers. More details of the two model configurations are presented in Tab. 8.

We use AdamW (Loshchilov & Hutter, 2017) ($\beta_1 = 0.9, \beta_2 = 0.95$) optimizer with weight decay 0.05 for model optimization. The initial learning rate is zero, which is linearly warmed up to $4\times10^{-4}$

for the first 3k steps and then decay to zero by cosine scheduling. Training this 'Ours (L)' for 40 epochs takes 128 Nvidia A100 GPUs for about one week.

We use the following techniques to save the GPU memory for our model training: 1) Mixed precision with BFloat16, 2) deferred back-propagation in NeRF rendering (Zhang et al., 2022), and 3) gradient checkpointing at every 4 self-attention layers. We also adopt the FlashAttention V2 (Dao, 2023) to reduce the overall training time.

## A.4 ADDITIONAL DETAILS OF EXPERIMENTAL SETTINGS

**Training datasets.** We render the Objaverse dataset following the same protocol as LRM (Hong et al., 2023) and DMV3D (Xu et al., 2023): each object is normalized to $[-1, 1]^3$ box and rendered at 32 random viewpoints. We also preprocess the MVImgNet captures to crop out objects, remove background [2], and normalizing object sizes in the same way as LRM and DMV3D. In total, we have multi-view images of ~1 million objects in our training set: ~730k from Objaverse, ~220k from MVImgNet.

**Evaluation datasets.** For OmniObject3D, GSO, ABO datasets, we render out 5 images from randomly selected viewpoints for each object; to ensure view sparsity, we make sure viewing angles between any two views are at least 45 degrees. We feed randomly-chosen 4 images to our model to predict a NeRF and poses, while using the remaining 1 to measure our novel-view rendering quality. For CO3D dataset, we note that CO3D captures may not cover the objects in 360°, hence we do not sparsify the input poses but instead use randomly selected views. For CO3D dataset, to remove background, we use the masks included in the CO3D dataset. However, we note that these masks can be very noisy sometimes, negatively affecting our model's performance and the baseline RelPose++ (mask variant). We randomly select 4 random input views for each capture. For DTU dataset, we take the 15 objects with manually annotated masks provided by IDR (Yariv et al., 2020); for each object, we randomly select 8 different combinations of four input views, resulting in a total of 120 different testing inputs.

**Evaluation baseline methods.** For the pose prediction task, as we evaluate the cross-dataset generalization capability of different methods, which reflects their performance when deployed in real-world applications. In this regard, we directly use the pretrained checkpoints from baselines, including FORGE (Jiang et al., 2022), HLoc (Sarlin et al., 2019) and RelPose++ (Lin et al., 2023a), for comparisons.

For the pose prediction task, as the pre-trained FORGE model expects input images to have black background, we replace the white background in our rendered images with a black one using the rendered alpha mask before feeding them into FORGE. RelPose++ has two variants: one trained on images with background (*w/ bg*) and the other trained on images without background (*w/o bg*). We evaluate the *w/o bg* variant on these datasets featuring non-informative white background.

For the novel view synthesis comparison with FORGE, note that we actually feed images with black background into FORGE, and evaluate PSNR using images with black background; this is, in fact, an evaluation setup that bias towards FORGE, as images with black background tend to have higher PSNR than those with white background.

## A.5 ADDITIONAL RESULTS

Table 9: Category-level comparison of pose prediction results with baseline RelPose++ (Lin et al., 2023a) on CO3D dataset (Reizenstein et al., 2021). We report the mean pose errors and (top two rows) and rotation accuracy@15° (bottom two rows) on 10 different test categories.

| | Ball | Book | Couch | Fris. | Hot. | Kite | Rem. | Sand. | Skate. | Suit. | Mean |
|---|---|---|---|---|---|---|---|---|---|---|---|
| RelPose++ (w/ bg) | 30.29 | 31.34 | **24.82** | 34.01 | **21.61** | 50.18 | 32.00 | 30.84 | 36.91 | 14.13 | 31.85 |
| Ours | **17.17** | **8.36** | 29.04 | **20.16** | 27.88 | **16.18** | **6.05** | **15.92** | **25.39** | **12.03** | **17.82** |
| RelPose++ (w/ bg) | 0.613 | 0.782 | **0.787** | 0.742 | **0.742** | 0.570 | 0.767 | 0.697 | 0.630 | 0.893 | 0.723 |
| Ours | **0.787** | **0.947** | 0.688 | **0.780** | 0.697 | **0.807** | **0.944** | **0.890** | **0.778** | **0.923** | **0.824** |

---

[2]Mask removal tool: https://github.com/danielgatis/rembg

**Category-level results on CO3D dataset.** In Tab. 9 we report the per-category results and comparisons to RelPose++ on held-out CO3D test set provided by RelPose++ Lin et al. (2023a). We outperform RelPose++ (*w/ bg*) on 8 out of 10 categories, despite that we are not trained on CO3D training set while RelPose++ is. In addition, our model is now limited to handle images without background; hence we use the masks included in the CO3D dataset to remove background before testing our model. The masks, however, seem to be very noisy upon our manual inspection; this negatively influenced our model's performance, but not RelPose++ (*w/ bg*). An interesting future direction is to extend our model to support images with background to in order to lift the impacts of 2D mask errors.

Table 10: Evaluation results on GSO data (Downs et al., 2022) rendered by FORGE (Jiang et al., 2022). We note that these renderings are a bit darker than majority of our training images, but our model still generalizes well to this dataset. Our model produces sharper renderings than FORGE (indicated by the higher SSIM score), while producing more accurate camera estimates.

| Method | R. error | Acc.@15° | Acc.@30° | T. error | $PSNR_{g.t.}$ | $PSNR_{pred.}$ | $SSIM_{g.t.}$ | $SSIM_{pred.}$ |
|---|---|---|---|---|---|---|---|---|
| FORGE | 50.20 | 0.253 | 0.514 | 0.573 | 21.25 | 22.90 | 0.767 | 0.793 |
| FORGE (refine) | 49.02 | 0.307 | 0.527 | 0.548 | 22.08 | **25.89** | 0.767 | 0.838 |
| Ours | **8.37** | **0.908** | **0.954** | **0.105** | **23.05** | 24.42 | **0.860** | **0.886** |

A.6    ADDITIONAL CROSS-DATASET EVALUATIONS

To further demonstrate the generalization capability of our model, we evaluate our model (trained on a mixture of Objaverse and MVImgNet) on another version of GSO dataset (Downs et al., 2022) (which is rendered by the FORGE paper). Note that these renderings are a bit darker than majority of our training images, but as shown in Tab. 1, our model still generalizes well to this dataset. Our model produces sharper renderings than FORGE with and without its per-scene optimization-based refinement (indicated by the higher SSIM score), while producing much more accurate camera estimates. Note that $PSNR_{g.t.}$, $SSIM_{g.t.}$ reflect how well renderings of our predicted NeRF using ground-truth input poses match the input images, while $PSNR_{pred}$, $SSIM_{pred}$ measures how well renderings of our predicted NeRF using ground-truth input poses match the inputs.

A.7    SCALING UP TRAINING OF RELPOSE++

To further demonstrate our method's superiority over the baseline method RelPose++ (Lin et al., 2023a), we re-train RelPose++ on the Objaverse dataset until full convergence for a more fair comparison. We then compare the re-trained model with our model ('Ours (S)' and 'Ours (L)') trained on exactly the same Objaverse renderings in Tab. 11. The re-trained RelPose++ using Objaverse does improve over the pretrained one using CO3D on the unseen test sets, OmniObject3D, GSO and ABO. However, our models (both 'Ours (S)' and 'Ours (L)') consistently outperform the re-trained baseline by a large margin in terms of rotation and translation prediction accuracy. We attribute this to our joint prediction of NeRF and poses that effectively exploit the synergy between these two tasks; in addition, unlike RelPose++ that regresses poses, we predict per-view coarse point cloud (supervised by distilling our predicted NeRF geometry in an online manner) and use a differentiable solver to get poses. This make us less prone to getting stuck in pose prediction local minimas than regression-based predictors, as also pointed out by Chen et al. (2022b).

A.8    APPLICATION

**Text/image-to-3D generation.** Since our model can reconstruct NeRF from 2-4 unposed images, it can be readily used in downstream text-to-3D applications to build highly efficient two-stage 3D generation pipelines. In the first stage, one can use geometry-free multi-view image generators, e.g., MVDream (Shi et al., 2023b), Instant3D (Li et al., 2023), to generate a few images from a user-provided text prompt. Then the unposed generated images can be instantly lifted into 3D by our *PF-LRM* with a single feed-forward inference (see Fig. 1). Or alternatively, one can generate a single image from text prompts using Stable Diffusion (Rombach et al., 2022), feed the single image to image-conditioned generators, e.g., Zero-1-to-3 (Liu et al., 2023a), Zero123++ (Shi et al., 2023a), to generate at least one additional view, then reconstruct a NeRF from the multiple unposed images

Table 11: Comparisons of cross-dataset generalization on GSO (Downs et al., 2022), ABO (Collins et al., 2022), OmniObject3D (Wu et al., 2023) with RelPose++ (Lin et al., 2023a) using the author-provided checkpoint (trained on CO3D (Reizenstein et al., 2021) and our **re-trained** checkpoint (trained on Objaverse (Deitke et al., 2023)). 'Ours (S)' and 'Ours (L)' are trained only on Objaverse as well for fair comparison. Though the **re-trained** RelPose++ improves over the pretrained version, we (both 'Ours (S)' and 'Ours (L)') still achieve much better pose prediction accuracy than it.

| Dataset | Method | R. error ↓ | Acc.@15° ↑ | Acc.@30° ↑ | T. error ↓ |
|---|---|---|---|---|---|
| *OmniObject3D* | RelPose++ (w/o bg, pretrained) | 69.22 | 0.070 | 0.273 | 0.712 |
| | RelPose++ (w/o bg, Objaverse) | 58.67 | 0.304 | 0.482 | 0.556 |
| | Ours (S) | 15.06 | 0.695 | 0.910 | 0.162 |
| | Ours (L) | **7.25** | **0.958** | **0.976** | **0.075** |
| *GSO* | RelPose++ (w/o bg, pretrained) | 107.49 | 0.037 | 0.098 | 1.143 |
| | RelPose++ (w/o bg, Objaverse) | 45.58 | 0.600 | 0.686 | 0.407 |
| | Ours (S) | 13.08 | 0.848 | 0.916 | 0.135 |
| | Ours (L) | **2.46** | **0.976** | **0.985** | **0.026** |
| *ABO* | RelPose++ (w/o bg, pretrained) | 102.30 | 0.060 | 0.144 | 1.103 |
| | RelPose++ (w/o bg, Objaverse) | 45.39 | 0.693 | 0.708 | 0.395 |
| | Ours (S) | 26.31 | 0.785 | 0.822 | 0.249 |
| | Ours (L) | **13.99** | **0.883** | **0.892** | **0.131** |

using our *PF-LRM*. In the latter approach, we can have a feed-forward single-image-to-3D pipeline as well, if the text-to-image step is skipped, as shown in Fig. 1.

## A.9 LIMITATIONS

Despite the impressive reconstruction and pose prediction performance of our model, there are a few limitations to be addressed in future works: 1) First, we ignore the background information that might contain rich cues about camera poses, e.g., vanishing points, casting shadows, etc, while predicting camera poses. It will be interesting to extend our work to handle background with spatial warpings as in (Zhang et al., 2020; Barron et al., 2022). 2) Second, we are also not able to model view-dependent effects due to our modelling choice of per-point colors, compared with NeRF (Mildenhall et al., 2020; Verbin et al., 2022). Future work will include recovering view-dependent appearance from sparse views. 3) The resolution of our predicted triplane NeRF can also be further increased by exploring techniques like coarse-to-fine modelling or other high-capacity compact representations, e.g., multi-resolution hashgrid (Müller et al., 2022), to enable more detailed geometry and texture reconstructions. 4) Our model currently assumes known intrinsics (see Sec. 3.1) from the camera sensor metadata or a reasonable user guess; future work can explore techniques to predict camera intrinscis as well. 5) Although our model is pose-free during test time, it still requires ground-truth pose supervision to train; an intriguing direction is to lift the camera pose requirement during training in order to consume massive in-the-wild video training data.

