# OpenReview forum: "PF-LRM: Pose-Free Large Reconstruction Model for Joint Pose and Shape Prediction"
_ICLR.cc/2024/Conference — ICLR 2024 spotlight_

### Official Review · Reviewer_XwFE · 2023-10-29

**Soundness:** 2 fair
**Presentation:** 3 good
**Contribution:** 3 good
**Rating:** 8
**Confidence:** 5

**Summary:**

This paper proposes a novel method for lifting a few unposed images directly into a 3D representation as well as estimating the corresponding camera poses. The key contributions are a novel object-centric pose estimation pipeline based on the Perspective-n-Point (PnP) algorithm, scaling up training to achieve strong cross-dataset generalization, and instant estimation of the underlying NeRF (as opposed to training offline via gradient descent). The authors demonstrate impressive results on object-centric scenes across datasets.

**Strengths:**

-The motivation for the method is clear - reconstructing object-centric 3D scenes without knowledge of the camera poses.
-The method is simple (which is a good thing) and straightforwardly scalable.
-The approach for camera pose estimation is neat, containing an attractive 3D inductive bias and mostly outperforms common object-centric pose estimation methods.
-Generated results are strong and aesthetically pleasing, with significant cross-dataset generalization demonstrated.
-Ablations are informative.

**Weaknesses:**

-I have a suspicion that the baselines - RelPose++ and Forge - are not trained on the same datasets, but that the authors are rather using the pre-trained methods and just evaluate them on these datasets. Could the authors clarify?
- One baseline is very clearly missing - the Scene Representation Transformer, Sajjadi et al, which can similarly perform novel view synthesis from unposed images at test time while requiring camera poses at training time.
- The prior point is even more critical since the authors don't always outperform RelPose++ on Co3D. If RelPose++ wasn't trained at a similar scale, this is concerning.
- In the evaluation section, the authors state that they evaluate PSNR against the input images. That is questionable - they should evaluate the PSNR against held-out test views?
- The authors should clarify - already in the introduction and the abstract - that their method still requires ground-truth poses at training time. I.e., it is only pose-free at test time!
- Why do the authors choose to report results on Co3D in the appendix as opposed to the main paper? I don't see a principled difference here. Just as the other results, this result should be reported in the main paper. The fact that their method requires training on an additional real-world dataset is important and relevant and should not be hidden in the appendix.
- The authors should demonstrate their method on some real-world captures instead on only synthetic examples.

Relevant references missed:
- Triplanes for neural fields were first proposed in "Convolutional Occupancy Networks", Peng et al. and should be discussed.
- The paper "FlowCam" also incorporates 3D shape information during the camera pose prediction process and should be discussed.

Minor concerns:
- In all the tables, the best-performing method should be bolded.
- "output-performing" --> outperforming (last paragraph before related work)
- I would recommend to avoid phrases like “pretty well” - sounds a bit casual.

**Questions:**

See weaknesses - in summary, I think this paper could be strong, but if the authors want to claim that their method outperforms prior pose-estimation methods, then they have to provide a fair benchmark and train these other methods on the same datasets and with a comparable level of compute as well, or at least provide a clear ablation study where they uncover that their model scales better with data than prior methods.

If the authors address this issue, I'm happy to raise my score significantly.

---

> ### Author Response · Authors · 2023-11-13
>
> We thank the reviewers for the valuable feedback. We will include the missing references and fix typos in a revision. We address your major concerns below.
>
> **Evaluating RelPose++ and Forge**: we used the pretrained checkpoints from RelPose++ and Forge to evaluate their cross-dataset generalization. We would like to emphasize that we focus on cross-dataset generalization in this work as it is vital for practical applications; our model was not trained on any of the evaluation datasets either. Moreover, we would like to highlight that we also reconstruct NeRF while RelPose++ doesn’t. This joint prediction of 3D and pose is very helpful for boosting pose prediction accuracy in our experiments, as shown in Tab. 4.
>
> **Missing Scene representation transformer (SRT) baseline**:  In our humble opinion, we can not consider SRT as a fair baseline to be compared, since there’s a clear distinction in objectives and scopes between our work and SRT. PF-LRM targets 3D reconstruction and pose estimation, while SRT only focuses on novel view synthesis. It means that SRT does not estimate poses for input images which is one of the main goals of this paper and densely measured. Poses are important for practical applications like robotic navigation, etc. Also, geometry can be naturally extracted from our reconstructed NeRF, while SRT is a geometry-free method that does not directly predict geometry. However, we thank the reviewer for bringing this work to our attention, we will cite SRT and discuss it in a revision.
>
> **Input-view PSNR**: We will include novel-view psnr in a revised draft before the rebuttal deadline.  The current input-view psnr is to show that our predictions (both NeRF and poses) are well-aligned with the input images, as we render our predicted NeRF using our predicted poses (in Fig. 3); we would like to highlight that we do well in this important objective of matching re-renderings with inputs for joint 3D and pose prediction, while baselines struggle with this goal.  We also show extracted geometry in Fig. 3, 4 and interactive NeRF viewers on our project page to demonstrate that we did not cheat by predicting degenerate solutions. In Fig. 4, we show re-rendering of our reconstructed NeRF using ground-truth pose, to demonstrate that removing pose prediction from our model leads to lower reconstruction quality. This is to highlight the synergy between NeRF reconstruction and pose estimation, which was not shown in prior work like SRT.
>
> **Comparison with RelPose++ on Co3D**: as shown in Tab. 8 of the appendix, we outperform RelPose++ on the majority of the Co3D categories, even though our model is not trained on Co3D while RelPose++ was trained on it. As to why we are worse on certain categories, we attribute this to the noisy in-accurate masks provided by Co3D, which we used to remove background before running our method. A clear performance drop can also be seen in Tab. 8 when we test RelPose++ on such background-free images. We think handling background in our framework is an interesting future direction.
>
> **Move Co3D results to the main paper**: We think it’s a good idea to move Co3D results to the main paper. By the time of submission, we did not have time and resources to train our model to full convergence on a mix of Objavese and MvImgNet data; hence we put the non-convergent model’s results in the appendix. We will include the results of our convergent model in the main paper in a revision.
>
> **Missing real-data results in the paper**: We respectfully disagree with this claim. We have shown our model’s results on the real DTU dataset (Fig 5 and Tab 9 in appendix), OmniObject3D dataset (Tab 1), Co3D dataset (Tab 8 in appendix). Please refer to our main paper and appendix for details.

---

> > ### Author Response · Authors · 2023-11-19
> >
> > Dear reviewer, please kindly let us know if our response has addressed your concerns with the approach of the rebuttal deadline. We are happy to answer any of your remaining concerns if you have any. We would also appreciate your response regarding whether you might be willing to raise your score. Thank you!

---

> ### Comment · Reviewer_XwFE · 2023-11-20
> **Reply**
>
> Dear Authors,
>
> thank you for the kind and detailed reply.
>
> ## Comparison to RelPose
>
> I do not think that your argument for evaluating with the pre-trained checkpoints for RelPose is valid. If I understand correctly, you are basically saying "we are evaluating cross-dataset generalization, therefore, it is reasonable to evaluate the pre-trained checkpoint for RelPose.". I don't think that is a valid argument at all. You are *claiming* that your *method* outperforms RelPose on pose prediction, as you literally say in your abstract:
>
> "When trained on a huge amount of multiview data, PF-LRM shows strong cross-dataset generalization ability, and outperforms baseline methods by a large margin in terms of pose prediction accuracy and 3D reconstruction quality on various evaluation datasets".
>
> If you want to claim that you outperform either of these methods on pose estimation accuracy, then you have two options: 1) you train your method on the *same* dataset as RelPose and then show that you are doing better or 2) you are training RelPose on the same dataset that you used for training and then show that you do better on the corresponding test set.
>
> Otherwise, you have not supported your claim. In other words, you would be putting out a paper that might lead many scientists down the wrong path: You are suggesting that anyone who is doing pose estimation ought to use your method, when in practice, it might well be that RelPose would in fact outperform your method when trained on the same dataset.
>
> IMO, it is standard procedure in our field to train the baseline on the same dataset as the proposed method to allow for a fair comparison.
>
> ## Other points
> Thank you for clarifying the other points, they clarified my other questions!

---

> > ### Comment · Reviewer_XwFE · 2023-11-22
> > **Follow-up**
> >
> > I have conferred with the other reviewers and got their opinion on this topic, and they similarly think that the comparison, as done, does not support the claims of the paper.
> >
> > However, I also think that this should not be in the way of this paper, which I think all the reviewers agree is good & novel!
> >
> > However, I would ask the authors to add several disclaimers to their paper about the RelPose comparison. Specifically, I would ask the authors to (1) add a "Limitations" section and (2) add a sentence to the experiments section and (3) add a sentence to the introduction section where they respectively point out that RelPose++ or Forge might perform better if trained on a larger-scale dataset, that this has not been attempted in this paper, and that hence, it is not clear whether the performance stems from a methods improvement or from the large-scale training, and that that is left to future work.
> >
> > Other than that, I have increased my score to "5", since I believe this paper is good and warrants publication. I will increase my score to "7" if the authors are willing to address this shortcoming, and will happily argue for acceptance.

---

> ### Author Response · Authors · 2023-11-22
> **Official Comment by Authors**
>
> We sincerely understand your concerns of fair comparison with baselines. We spent the rebuttal period scaling up the training of the baseline RelPose++ on the Objaverse dataset, and report the performance of the re-trained baseline in the following table. We can see that our method (trained on exactly the same data) still outperforms the retrained baseline by a wide margin, demonstrating the superiority of our proposed method. We also attempted to scale up the training of baseline FORGE, but its complex multi-stage training (6 stages) seems non-trivial to scale up in our trials. We leave it to future work to investigate the scalability issue of FORGE.
>
> ---
>
> OmniObject3D
>
> | Method                                	| R. error | Acc.@15   | Acc.@30   | T. error  |
> |-------------------------------------------|----------|-----------|-----------|-----------|
> | RelPose++ (w/o bg, pretrained)        	| 69.22	| 0.070 	| 0.273 	| 0.712 	|
> | RelPose++ (w/o bg, trained on Objaverse)  | 58.67	| 0.304 	| 0.482 	| 0.556 	|
> | Ours (S)                              	| 15.06	| 0.695 	| 0.910 	| 0.162 	|
> | Ours (L)                              	| **7.25** | **0.958** | **0.976** | **0.075** |
>
> ---
>
> GSO
>
> | Method                                	| R. error | Acc.@15  | Acc.@30  | T. error  |
> |-------------------------------------------|----------|----------|----------|-----------|
> | RelPose++ (w/o bg, pretrained)        	| 107.49   | 0.037	| 0.098	| 1.143 	|
> | RelPose++ (w/o bg, trained on Objaverse)  | 45.58	| 0.600	| 0.686	| 0.407 	|
> | Ours (S)                              	| 13.08	| 0.848	| 0.916	| 0.135 	|
> | Ours (L)                              	| **2.46** | **0.976**| **0.985**| **0.026** |
>
> ---
>
> ABO
>
> | Method                                	| R. error  | Acc.@15   | Acc.@30   | T. error  |
> |-------------------------------------------|-----------|-----------|-----------|-----------|
> | RelPose++ (w/o bg, pretrained)        	| 102.30	| 0.060 	| 0.144 	| 1.103 	|
> | RelPose++ (w/o bg, trained on Objaverse)  | 45.39 	| 0.693 	| 0.708 	| 0.395 	|
> | Ours (S)                              	| 26.31 	| 0.785 	| 0.822 	| 0.249 	|
> | Ours (L)                              	| **13.99** | **0.883** | **0.892** | **0.131** |
>
> ---
>
> Updated draft: we have updated (in-place) the abstract, introduction, related work, conclusion sections. We’ve mentioned in the updated abstract that we used multi-view posed data for training. We’ve cited SRT paper and discussed the scope difference between our work and SRT. We also added citations to the Convolutional Occupancy Networks paper and FlowCam paper, as the reviewer suggested. For the experiment section, we did not modify it in-place because our modifications will change the figure/table numbers, hence increasing the reading difficulty for the reviewers. Instead, we put the updated experiment section in Appendix B. We plan to integrate it upon acceptance. We are also happy to integrate it now, if the reviewers don’t think this increases reading burden. You can find our novel-view PSNR in Tab. 14 and our novel-view renderings (along with ground-truth) in Fig. 9. We also include a detailed limitation section in Appendix C.

---

> > ### Comment · Reviewer_XwFE · 2023-11-22
> > **Reply**
> >
> > This is great - thank you very much for being receptive to this feedback, I think that these experiments serve as a great reference and really support your claim!
> >
> > I increased my score to 8!

---

### Official Review · Reviewer_T3us · 2023-10-30

**Soundness:** 3 good
**Presentation:** 3 good
**Contribution:** 4 excellent
**Rating:** 8
**Confidence:** 4

**Summary:**

The paper proposes a 3D reconstruction method that focuses on shape and pose predictions given a sparse set of un-posed input images. The proposed method consists of two major components: 1) a transformer that aggregates DINO-based image tokens from multi-view images and 2) a DSAC-fashion pose estimator that predicts 3D points coordinates from aggregated features followed by a differentiable PnP refinement. While aggregating multi-view features, the transformer also enables shape reconstruction by predicting NeRF parameters in terms of Triplane features.

**Strengths:**

1. The paper is generally well written and easy to follow.
2. The method looks convincing and the paper is addressing a critical challenge in this field.
3. The anonymous website provides extra demos.

**Weaknesses:**

1. In Figure 3 and Figure 4, I am confused about why the results are comparing input images with their predictions? Shouldn’t it be comparing GT novel view images with rendered novel views, as in Figure 5? Given input images, it’s not surprising that the reconstruction of input images should be almost perfect right?
2. The first issue leads to further confusion when reading Figure 4. I am not sure I understand correctly, but it seems like pose prediction makes limited difference.
3. Missing related works. Section 3.3 is about pose predictions and PnP. This approach is closely related to the line of work in *scene coordinate regression* [a][b], which is not mentioned at all in the related work section.

**Reference**

[a] Shotton, Jamie, Ben Glocker, Christopher Zach, Shahram Izadi, Antonio Criminisi, and Andrew Fitzgibbon. "Scene coordinate regression forests for camera relocalization in RGB-D images." In Proceedings of the IEEE conference on computer vision and pattern recognition, pp. 2930-2937. 2013.

[b] Brachmann, Eric, Alexander Krull, Sebastian Nowozin, Jamie Shotton, Frank Michel, Stefan Gumhold, and Carsten Rother. "Dsac-differentiable ransac for camera localization." In Proceedings of the IEEE conference on computer vision and pattern recognition, pp. 6684-6692. 2017.

**Questions:**

Refer to points 1 and 2 in the "Weaknesses" section.

While the paper presents a novel idea and convincing quantitative results, these two points create confusions and prevent me from making a definitive evaluation. Once these issues are resolved, I am willing to increase my evaluation to accept.

---

> ### Author Response · Authors · 2023-11-13
>
> We thank the reviewer for the valuable feedback. We will add the suggested references ([a],[b]) in a revised draft soon before the rebuttal deadline. Please find our answers to your questions below.
>
> **Input-view PSNR**: We will include novel-view psnr in a revised draft before the rebuttal deadline.  The current input-view PSNR is to show that our predictions (both NeRF and poses) are well-aligned with the input images, as we render our predicted NeRF using our predicted poses; we would like to highlight that we do well in this important objective of matching re-renderings with inputs for joint 3D and pose prediction, while baselines struggle with this goal. We also show extracted geometry in Fig. 3, 4 and interactive NeRF viewers on our project page to demonstrate that we did not cheat by predicting degenerate solutions.

---

> > ### Author Response · Authors · 2023-11-19
> >
> > Dear reviewer, please kindly let us know if our response has addressed your concerns. We are happy to answer any of your remaining concerns and questions if you have any. We would also appreciate your response regarding whether you might be willing to raise your score. Thank you!

---

> > ### Comment · Reviewer_T3us · 2023-11-21
> > **No further questions**
> >
> > Dear authors,
> >
> > Thanks for the detailed response. I don't have further questions about the content, but it seems like there is no revised draft updated at this moment?
> >
> > Best,
> > T3us

---

> > > ### Author Response · Authors · 2023-11-22
> > > **Official Comment by Authors**
> > >
> > > Dear reviewer, thank you for your reminder. We have updated **(in-place)** the abstract, introduction, related work, conclusion sections. We have added citations to the suggested references. For the experiment section, we did not modify it in-place because our modifications will change the figure/table numbers, hence increasing the reading difficulty for the reviewers. Instead, we put the updated experiment section in Appendix B. We plan to integrate it upon acceptance. We are also happy to integrate it now, if the reviewers don’t think this increases reading burden. You can find our novel-view PSNR in Tab. 14 and our novel-view renderings (along with ground-truth) in Fig. 9.

---

> > > > ### Comment · Reviewer_T3us · 2023-11-22
> > > >
> > > > Thanks for updating the draft with the new tables and figures. These results have largely addressed my concerns. I have updated my rating to accept.

---

### Official Review · Reviewer_Pk6q · 2023-10-31

**Soundness:** 4 excellent
**Presentation:** 3 good
**Contribution:** 4 excellent
**Rating:** 8
**Confidence:** 3

**Summary:**

This paper proposes a method to reconstruct 3d models from unposed sparse-view images. To enable this goal, the authors use a single-stream transformer to simultaneously process image patch tokens and nerf tokens, resulting in a simultaneously estimation of triplane-nerf and camera poses. The estimation of camera poses utilizes predicted per-view coarse geometry and differentiable pnp. This method shows strong cross-dataset generalization ability and outperforms baselines on various datasets.

**Strengths:**

1. The authors conducted rich experiments to prove the effectiveness of their method.
2. Reconstructing 3d models from sparse-view images is an important topic in the AIGC era.
3. Employing differentiable pnp into nerf reconstruction is a good idea to handle camera pose uncentainty.

**Weaknesses:**

The authors mainly handled the cases of 2-4 views (e.g. Figure 1). Is there any criteria for the authors to select image views? It seems that the tested camera views have suitable (or informative) relative angles (about 30 degrees to about 120 degrees). What would happen if extreme/degenerated views are provided (e.g. the front and back views that are parallel, or views with small angles, or views with translational displacement only)?   I think explaing this would make the study stronger.

**Questions:**

No further questions. I think the paper provides enough  details.

---

> ### Author Response · Authors · 2023-11-13
>
> We thank the reviewer for the positive feedback about our empirical evaluation, technical soundness, and potential impact on the 3D generative AI. We address your questions below.
>
> **View selection**: For the evaluation on ABO and GSO dataset, we randomly select 4 sparse views such that viewing angles between any two views are at least 45 degrees to ensure sparsity. This is to test our method’s test-time performance on extreme inputs. But our method can work on different viewpoint settings, as during training, we did not enforce any restriction in selecting input viewpoints. The bunny and frog in Fig. 1 shows that our method can work well even when the two input views have little overlap. Our model can also handle the easier case where viewpoint changes are small.

---

> > ### Comment · Reviewer_Pk6q · 2023-11-19
> > **Comments for the feedback**
> >
> > Thanks for the feedback! It solve my concerns. However, after reading all the comments of other reviewers, I notice that I missed a truth that the proposed method "still requires ground-truth poses at training time" (as reviewer XwFE  pointed out). This fact lowers my rating of this paper. But I still think the application of differentiable pnp solver in this task is interesting.

---

> > > ### Author Response · Authors · 2023-11-19
> > >
> > > We thank the reviewer for the response. However, we would like to emphasize that applying ground-truth poses for supervision has been very standard in previous works, including state-of-the-art baseline methods we compare against such as RelPose++ and FORGE. Such ground-truth poses are readily available in both large-scale 3D synthetic dataset (e.g. Objaverse) and real-world dataset (e.g. MVImgNet, ground-truth poses are estimated by SfM), and therefore the usage of ground-truth poses do not introduce any extra burden or hinder the applicability of our method.
> > >
> > > Despite the aforementioned factors, we would like to point out that jointly estimating poses and 3D shapes from sparse unposed images is HIGHLY challenging, non-trivial and unresolved in the community. We have made the following novel contributions to achieve state-of-the-art performance:
> > > * We have introduced a novel single-stream transformer architecture (as opposed to the encoder-decoder architecture proposed in the LRM work). Such a network allows us to exchange information between 2D image tokens and 3D triplane tokens for joint pose estimation and shape reconstruction. To the best of our knowledge, this is the first framework which utilizes a unified transformer to jointly pose estimation and shape prediction.
> > > * Instead of directly regressing camera poses, we propose to predict per-patch points and apply a differentiable PnP solver to get the camera poses. We supervise the prediction of our per-patch points with our predicted NeRF in an online distillation manner, which is also only possible with our novel joint pose and NeRF prediction framework. Such a design significantly improves the performance of our method (see Table 11 of the paper).
> > > * We train our model on large-scale dataset, and our highly-scalable and high-capacity model can effectively consume the large-scale data to achieve strong cross-dataset generalization ability and outperform baseline methods by a LARGE margin.
> > >
> > > Considering all above, we would like to argue that the usage of ground-truth camera poses during training (our test-time inference is pose-free) does not hurt the contributions of our method. We hope the reviewer provides more clarifications about lowering the score and we would be happy to answer any questions the reviewer may have.

---

> > > ### Author Response · Authors · 2023-11-22
> > > **Official Comment by Author**
> > >
> > > Dear reviewer, please let us know if you have other concerns. We are more than happy to address them. Thank you!

---

### Official Review · Reviewer_5PYC · 2023-11-05

**Soundness:** 3 good
**Presentation:** 4 excellent
**Contribution:** 3 good
**Rating:** 8
**Confidence:** 4

**Summary:**

The paper presents an amortized inference model for a fast NeRF reconstruction of 3D objects from a few source views. These sources are either obtained from a view-aware diffusion model such as zero123 or give a ground truth. The work trains a large transformer model i.e. 0.59B parameter model for fast (1.3s) inference on A100GPUs. The transformer model outputs coarse pointclouds as well as triplanes for differentiable PNP based pose estimation and NeRF reconstruction respectively. The model is trained on Objaverse dataset and shows generalization capability on other datasets such as ABO, GSO, Omni Objects3D etc.

**Strengths:**

The paper proposes a technically sound amortized inference approach for 3D object reconstruction in a fast manner from a few unposed images. The strnegths of this paper are as follows:

1. Clearly better quantitative results against relevant pose predictor baselines
2. Good ablations showing how various factors affect the model performance, especially mask noise
3. A good downstream application of the method showing text-to-3D results which is an important application for this work

**Weaknesses:**

Although the paper is nicely written and the experiments are good, I have the following questions:

1. How much of the improvement comes from the data? The pipeline is not that novel since all of which the authors have presented have been shown before, even the differentiable PnP solver comes from prior work. I would have liked to see scaling laws of the performance on the increased number of data points.

2. No comparison with sparse NeRF methods[1, 2, 3] is shown. Is this because they require accurate camera poses? I would have liked to see some comparison or discussion with these methods, some of them already show sparse novel view synthesis using triplane formulations[1]

3. Relating to point 1, how much does the network learn training data distribution? Is it possible that due to the massive scale of the objaverse dataset, some of the generalizable evaluation, might be in questions, since we don't know if the network has seen similar data before. Can the authors comment on that with some analysis/discussion?

[1] Irshad et al, NeO 360: Neural Fields for Sparse View Synthesis of Outdoor Scenes
[2] Niemeyer et al, RegNeRF: Regularizing Neural Radiance Fields for View Synthesis from Sparse Inputs
[3] Truong et al, SPARF: Neural Radiance Fields from Sparse and Noisy Poses

**Questions:**

Please see my questions in the weakness section above.

---

> ### Author Response · Authors · 2023-11-13
>
> We thank the reviewer for the valuable feedback. Please see our answers to your questions below.
>
> **Limited pipeline novelty**: We respectfully disagree with this point. We would like to reiterate the contributions of our work: first, we propose a novel single-stream transformer architecture for joint NeRF and pose prediction, which is different from the encoder-decoder one used in the LRM, Instant3D work; the single-stream architecture not only allows the NeRF tokens to be contextualized by image tokens for reconstruction purpose, but  also allows the image tokens to be contextualized by NeRF tokens for predicting per-view coarse geometry used for solving poses. Second,  as far as we know, we are the first to apply the differentiable PnP solver to the task of jointly predicting 3D and pose from unposed sparse views, though the solver itself is inspired by the EPro-PNP work. This is in stark contrast to most prior work, e.g., RelPose++, Forge, that uses direct pose regression from images. Meanwhile, unlike the 6DoF pose estimation and 3D object detection tasks shown in EPro-PNP, we don’t have access to ground-truth 3D points/models during training (we only use multi-view posed images as training data); hence we propose the novel solution of supervising the sparse point predictions by distilling the predicted NeRF geometry in an online manner.
>
> **Overfitting concern**:  As shown by Fig. 1, we have tested our method on the generated multi-views from Zero1-to-3, MVDream, SyncDreamer etc. These images are unlikely to be in the Objaverse dataset, as they are generated by 2D diffusion models, but our method still works well on them, showcasing our model’s strong cross-dataset generalizability. Moreover, we’ve also demonstrated our method’s generalization to the real scanned datasets OmniObjects3D, as shown in Fig. 3 and Tab. 1. This dataset was published later than Objaverse, making it unlikely to be covered by Objaverse.
>
> **Scaling laws**: we have shown scaling laws for model sizes in Table 1,2,3 and 4, where bigger models clearly outperform smaller models. Due to limited compute resources, we do not show scaling laws for data sizes using our largest model (as the scaling law of data is not insightful under smaller model and with inadequate compute [1]); but we’re happy to include one in the final draft.     [1] Scaling Laws for Neural Language Models
>
> **Comparison with sparse NeRF methods**:  In this work, we focus on the challenging case of sparse unposed images as inputs, for the practical concern of estimating poses from sparse unposed images is a very challenging task itself. Hence, most sparse NeRF methods (e.g., [1,2,3] as the reviewer pointed out) requiring poses cannot work on such unposed inputs, especially when overlapping between input views is minimized (e.g., the frog and bunny in our Fig. 1).

---

> > ### Author Response · Authors · 2023-11-19
> >
> > Dear reviewer, please kindly let us know if our response has addressed your concerns. We are happy to answer any of your remaining concerns and questions if you have any. We would also appreciate your response regarding whether you might be willing to raise your score. Thank you!

---

> > > ### Comment · Reviewer_5PYC · 2023-11-20
> > > **Response to author's rebutal**
> > >
> > > Thank you for answering my questions. I still think the authors need to mention a discussion to sparse-view view synthesis literature i.e. [1,2,3] in their work, especially the ones that utilize triplanes for example [1] and others to highlight how they are lacking and the author's approach is superior.
> > >
> > > I genuinely think the work is a nice contribution, however after reading other reviewer's responses and discussion, i believe the work is missing one key comparison which is comparison to relpose++ and forge on the same dataset the authors trained on. This is to ensure an apples-to-apples comparison. It looks like high-quality datasets are really the key and if the authors are claiming their approach is superior, they should not be handicapping other approaches with limited datasets and expect that they would generalize better than them. I am decreasing my score one level due to these missing comparison and my comments above.

---

> > > > ### Author Response · Authors · 2023-11-22
> > > > **Official Comment by Authors**
> > > >
> > > > We sincerely understand your concerns of fair comparison with baselines. We spent the rebuttal period scaling up the training of the baseline RelPose++ on the Objaverse dataset, and report the performance of the re-trained baseline in the following table. We can see that our method (trained on exactly the same data) still outperforms the retrained baseline by a wide margin, demonstrating the superiority of our proposed method. We also attempted to scale up the training of baseline FORGE, but its complex multi-stage training (6 stages) seems non-trivial to scale up in our trials. We leave it to future work to investigate the scalability issue of FORGE.
> > > >
> > > > ---
> > > >
> > > > OmniObject3D
> > > >
> > > > | Method                                	| R. error | Acc.@15   | Acc.@30   | T. error  |
> > > > |-------------------------------------------|----------|-----------|-----------|-----------|
> > > > | RelPose++ (w/o bg, pretrained)        	| 69.22	| 0.070 	| 0.273 	| 0.712 	|
> > > > | RelPose++ (w/o bg, trained on Objaverse)  | 58.67	| 0.304 	| 0.482 	| 0.556 	|
> > > > | Ours (S)                              	| 15.06	| 0.695 	| 0.910 	| 0.162 	|
> > > > | Ours (L)                              	| **7.25** | **0.958** | **0.976** | **0.075** |
> > > >
> > > > ---
> > > >
> > > > GSO
> > > >
> > > > | Method                                	| R. error | Acc.@15  | Acc.@30  | T. error  |
> > > > |-------------------------------------------|----------|----------|----------|-----------|
> > > > | RelPose++ (w/o bg, pretrained)        	| 107.49   | 0.037	| 0.098	| 1.143 	|
> > > > | RelPose++ (w/o bg, trained on Objaverse)  | 45.58	| 0.600	| 0.686	| 0.407 	|
> > > > | Ours (S)                              	| 13.08	| 0.848	| 0.916	| 0.135 	|
> > > > | Ours (L)                              	| **2.46** | **0.976**| **0.985**| **0.026** |
> > > >
> > > > ---
> > > >
> > > > ABO
> > > >
> > > > | Method                                	| R. error  | Acc.@15   | Acc.@30   | T. error  |
> > > > |-------------------------------------------|-----------|-----------|-----------|-----------|
> > > > | RelPose++ (w/o bg, pretrained)        	| 102.30	| 0.060 	| 0.144 	| 1.103 	|
> > > > | RelPose++ (w/o bg, trained on Objaverse)  | 45.39 	| 0.693 	| 0.708 	| 0.395 	|
> > > > | Ours (S)                              	| 26.31 	| 0.785 	| 0.822 	| 0.249 	|
> > > > | Ours (L)                              	| **13.99** | **0.883** | **0.892** | **0.131** |
> > > >
> > > > ---
> > > >
> > > >
> > > > Updated draft: we have updated (in-place) the abstract, introduction, related work, conclusion sections. Detailed discussions with sparse NeRF methods requiring poses are included in the related work. For the experiment section, we did not modify it in-place because our modifications will change the figure/table numbers, hence increasing the reading difficulty for the reviewers. Instead, we put the updated experiment section in Appendix B. We plan to integrate it upon acceptance. We are also happy to integrate it now, if the reviewers don’t think this increases reading burden.

---

> > > > > ### Comment · Reviewer_5PYC · 2023-11-22
> > > > > **response resolved my concerns, I am willing to increase my score**
> > > > >
> > > > > Thanks to the authors for including additional comparisons and discussion to related sparse view NeRF methods. It resolved all my concerns and I increased my score to accept.

---

### Meta-Review · Area_Chair_JH6m · 2023-12-07

**Metareview:**

The submission received positive reviews from all the reviewers. The reviewers recognize the novelty and simplicity of the method as well as the strong results demonstrated. Initially, reviewers have concerns about comparisons against prior methods (RelPose++), but they have been subsequently addressed in the revisions. After reading the paper, the reviewers' comments, the authors' rebuttal and the discussions, the AC agrees with the decision by the reviewers and recommends acceptance.

**Justification For Why Not Higher Score:**

I believe the paper should demonstrate applicability to a broader domain (e.g. real-world datasets that are more difficult) for it to be considered for oral presentation.

**Justification For Why Not Lower Score:**

3D modeling from sparse observations with unknown poses is a very challenging ill-posed problem. This paper presents a simple yet scalable method and demonstrated strong results. I believe this should be highlighted for the community.

---

### Decision · Program_Chairs · 2024-01-16

Accept (spotlight)